# Stress granule assembly impairs macrophage efferocytosis to aggravate allergic rhinitis in mice

Ye Zhou [1,4], Zixuan Yang [2,3,4], Yuanyuan Wang [1,4], Yue Dong [1,4], Tianyu Wang [2,4], Yunhui Li [1], Caiquan Liang [2], Yanfang Liu [1], Zhixuan Li [1], Shanrong Liu [1], Liangchen Gui [1], Yiwen Fan [1], Ting Lei [1], Kaiwei Jia [1], Liyuan Zhang [1], Mu Wang [1], Wen Nie [1], Long Chen [1], Mingrui Ma [1], Yanfeng Wu [1], Cuiping Zhong [3], Huanhai Liu [2] & Jin Hou [1] ✉

Cytoplasmic stress granules (SG) assemble in response to stress-induced translational arrest and are key signaling hubs orchestrating cell fate and regulating various physiological and pathological processes. However, the role of SG formation in the progression of allergic diseases is incompletely understood. Here, by analyzing the nasal tissues of allergic rhinitis (AR) mouse models and AR patients, we find that SGs assemble specifically in the macrophages within the nasal mucosa and promote AR progression by restraining the efferocytotic ability of macrophages, ultimately resulting in reduced Mres generation and IL-10 production. Mechanistically, intracellular m7G-modified *Lrp1* mRNA, encoding for a typical efferocytosis receptor, is transported by the m7G reader QKI7 into stress-induced SGs, where *Lrp1* mRNA is sequestered away from the translation machinery, ultimately resulting in reduced macrophage efferocytosis. Therefore, SG assembly impairs macrophage efferocytosis and aggravates AR, and the inhibition of SGs bears considerable potential in the targeted therapy.

Stress granule (SG), a typical type of cytoplasmic membraneless organelle, is assembled under various stress by aggregating RNA-binding proteins and untranslated mRNAs, which results in the arrest of protein translation, thus orchestrating stress response and cell fate[1,2]. SGs assemble through liquid-liquid phase separation (LLPS) arising from a core protein-RNA interaction network, and the central node of this network is Ras GTPase-activating protein-binding protein 1 (G3BP1), which functions as a scaffold that triggers RNA-dependent LLPS in response to the elevated uncoated mRNAs[1,2]. Recent advances suggest that SG formation during stress plays important roles the regulation of various physiological and pathological processes, such as antiviral immune response, neurodegenerative diseases, and tumor progression[3–5]. However, the potential role of SG assembly in the progression of upper respiratory diseases has not been demonstrated until now.

Nasal mucosa serves as the crucial but vulnerable barrier of upper respiratory system against environmental stimulators such as pathogens, allergens, and pollutions[6]. Among the upper respiratory diseases, allergic rhinitis (AR) represents a global health problem affecting 20% to 30% of the population, with burdensome symptoms including nasal congestion, itching, rhinorrhea, and sneezing[7,8]. Allergen-specific Th2-mediated type II immune response, IgE production, and eosinophil infiltration have been well-recognized for the development of nasal mucosa inflammation, increased vascular permeability, and mucus

[1]National Key Laboratory of Immunity and Inflammation, Second Military Medical University, Shanghai 200433, China. [2]Department of Otolaryngology-Head and Neck Surgery, Second Affiliated Hospital of Second Military Medical University, Shanghai 200003, China. [3]Department of Otolaryngology-Head and Neck Surgery, No. 940 Hospital of Joint Logistics Support Force of People's Liberation Army, Lanzhou 730000, China. [4]These authors contributed equally: Ye Zhou, Zixuan Yang, Yuanyuan Wang, Yue Dong, Tianyu Wang. ✉e-mail: houjin@immunol.org

production [9,10]. Besides, some innate immune cells, such as neutrophils and group 2 innate lymphoid cells (ILC2), play critical roles in AR progression through neutrophil extracellular traps (NET) and type II cytokines produced by ILC2s[11–13]. Nasal macrophages also participate in the development of AR, especially their polarization is implicated in the etiology[14]. Given that the nasal mucosa is frequently exposed to various external stressors, it is intriguing to investigate whether SGs are assembled in the mucosa and participate in the progression of AR.

The dying of immune cells displays "eat-me" signals like phosphatidylserine (PS) and calreticulin (CALR), which can be recognized by phagocytes through a variety of receptors including TYRO3 protein tyrosine kinase (TYRO3), AXL receptor tyrosine kinase (AXL), MER proto-oncogene tyrosine kinase (MerTK), T cell immunoglobulin and mucin domain containing 4 (TIMD4), and LDL receptor related protein 1 (LRP1)[15,16]. This process is termed as efferocytosis, which is characterized by the phagocytes clearing dead cells to prevent hyperinflammation and resolve immunopathology, thus presenting the immunotolerant response. The potential role of macrophage efferocytosis and its regulation in upper respiratory allergic disease especially AR are still elusive.

Here, we apply immunofluorescence and RNA-scope assay in the nasal mucosa of AR mouse models and patients, and find that SGs are assembled specifically in the macrophages within the nasal mucosa. SG formation is shown to restrain the efferocytotic ability of macrophages and suppress the clearance of apoptotic immune cells, thus promoting AR progression. By using *G3bp1* macrophage-specific knockout mice and SGs inhibitor in an AR mouse model, we elucidate the potential role of SGs assembly in AR progression and find that SGs assembly in macrophages results in impaired efferocytosis and exacerbated AR symptoms, through the translational suppression of m7G-modified mRNA of macrophage efferocytosis receptor *Lrp1*. Therefore, the development of inhibitors or intervention strategies targeting SG assembly in macrophages may be promising for the targeted therapy of AR patients.

## Results

### SGs are assembled in the macrophages of the nasal mucosa during AR

Respiratory mucosa, especially nasal mucosa, is an important immune barrier exposed all the time to various irritations, like pathogens, cold air, pollen, and house dust mites (HDM), which may lead to the stress response in mucosa cells. Therefore, we first examined SG formation by staining its assembly scaffold G3BP1 in the nasal mucosa, and found the assembly of SGs in the basal mucosa of nose in the OVA-induced AR mice (Fig. 1a and Supplementary Fig. 1a). Since the basal layer of the nasal mucosa is composed of epithelial cells and immune cells, we performed RNA-scope assay and found that SGs were assembled dominatingly in the CD45+ immune cells (Fig. 1b). Whereafter, SG assembly was predominantly found in the CD11b+ myeloid cells but not lymphocytes through immunofluorescence staining (Fig. 1c and Supplementary Fig. 1b, c). Next, we performed single-cell RNA sequencing (scRNA-seq) on the CD45+ cells sorted from the nasal mucosa of control and OVA-induced AR mice. Uniform manifold approximation and projection (UMAP) analysis identified a total of 12 cell clusters according to the previously defined markers (Fig. 1d, e and Supplementary Fig. 1d, e)[17], and neutrophils, T cells, macrophages, and plasma cells were predominantly found in the nasal mucosa (Supplementary Fig. 1f). Further analysis showed that macrophages displayed much higher *G3bp1* expression than neutrophils (Fig. 1f, g and Supplementary Fig. 1g), suggesting that SGs may be predominately assembled in macrophages. Furthermore, we separated primary peritoneal macrophages (PM), bone marrow-derived macrophages (BMDM), bone marrow neutrophils (BMN), and bone marrow-derived eosinophils (BMDE) from mice, and determined that *G3bp1* is predominantly expressed in macrophages and the macrophage cell lines RAW 264.7

and iBMDM (Fig. 1h, i). Meanwhile, nasal mucosa macrophages (NMM), nasal mucosa neutrophils (NMN), and nasal mucosa T cells (NMT) were isolated from HDM-induced AR mice, and it was also determined that *G3bp1* is predominantly expressed in macrophages within the nasal mucosa (Fig. 1j). Additionally, SG assembly could be activated by both oxidative stress (sodium arsenite, $NaAsO_2$) and allergen HDM in macrophages, through LLPS which was certified by fluorescence recovery after photobleaching (FRAP) assay (Supplementary Fig. 1h–k and Supplementary Movies 1-3). Overall, these findings suggest that SGs are assembled in the macrophages of nasal mucosa during AR.

### Macrophage SG assembly promotes AR progression

To dissect the role of SG assembly in macrophages, we crossed *G3bp1* floxed (*G3bp1f/f*) mice to *Lyz2-Cre* and obtained macrophage-specific *G3bp1* knockout (*G3bp1mac-/-*) mice (Supplementary Fig. 2a). G3BP1 expression was efficiently ablated in the macrophages of *G3bp1mac-/-* mice (Supplementary Fig. 2b,c). Immunofluorescence microscopy for another SG marker, endogenous eukaryotic initiation factor 4 G (eIF4G), revealed that *G3bp1mac-/-* attenuated SG formation in macrophages exposed to HDM (Supplementary Fig. 2d). Next, we challenged *G3bp1mac-/-* mice with two AR mouse models (OVA and HDM) (Supplementary Fig. 2e) and *G3bp1mac-/-* mice exhibited alleviated AR symptoms, including sneezing and scratching (Fig. 2a). Since *Lyz2-Cre* could be engaged in the myeloid cell lineage including monocytes, macrophages, and granulocytes, we further crossed *G3bp1f/f* mice to *Ly6G-Cre* and *Siglecf-Cre* to obtain neutrophil-specific *G3bp1* knockout (*G3bp1neu-/-*) and eosinophil-specific *G3bp1* knockout (*G3bp1eos-/-*) mice. Applied with the OVA-induced AR mouse model, *G3bp1neu-/-* and *G3bp1eos-/-* mice exhibited similar AR symptoms to controls (Supplementary Fig. 2f). Thus, SG assembly in macrophages, but not in neutrophils or eosinophils, aggravates AR symptoms. Simultaneously, histological examination also confirmed that macrophage *G3bp1* deficiency attenuated epithelial shedding, cell infiltration, and exudation in the nasal mucosa with lower histological score, and reduced mucus secretion with fewer PAS+ goblet cells compared to the control group (Fig. 2b, c and Supplementary Fig. 2g–j). Furthermore, lower infiltration of eosinophils, neutrophils, and T cells (Fig. 2d and Supplementary Fig. 2k), and lower cell infiltration in nasal lavage fluid (NLF) (Fig. 2e) in *G3bp1mac-/-* mice than those in controls were determined by flow cytometry analysis. However, there was no significant change of F4/80+ macrophages infiltration within nasal mucosa between *G3bp1f/f* and *G3bp1mac-/-* mice (Supplementary Fig. 2k), while the generation of inflammation resolution-associated macrophages (Mres, F4/80medCD11blow) was increased by *G3bp1mac-/-* (Fig. 2f), which is known for the control of inflammatory diseases[18,19]. Next, less IL-4, IL-5, IL-13, IL-6, Prg2, Epx, and Mpo expression in the nasal mucosa (Fig. 2g and Supplementary Fig. 2l), and lower IL-4, IL-5, and IL-13 production in NLF of *G3bp1mac-/-* mice than those of controls were determined in challenged mice and not their unchallenged controls (Fig. 2h). Otherwise, more IL-10 expression in the nasal mucosa and NMMs of *G3bp1mac-/-* mice than those of controls were determined in challenged mice (Fig. 2g, i and Supplementary Fig. 2l). Meanwhile, serum OVA-specific IgE concentration was also decreased in *G3bp1mac-/-* mice (Fig. 2j). Additionally, TUNEL staining determined that apoptotic cells were decreased in the nasal mucosa of *G3bp1mac-/-* mice in both OVA and HDM-induced AR models (Supplementary Fig. 2m, n). Since Th2 response is important for AR, we measured *Il-4* mRNA expression during Th2 differentiation, and found no significant difference between *G3bp1f/f* and *G3bp1mac-/-* (Supplementary Fig. 2o). Together, these results suggest that SG assembly within macrophages promotes AR progression.

### SG assembly in macrophages impairs efferocytosis

We further investigated the mechanism responsible for the macrophage SG assembly-promoted AR. BMDMs were isolated from *G3bp1f/f*

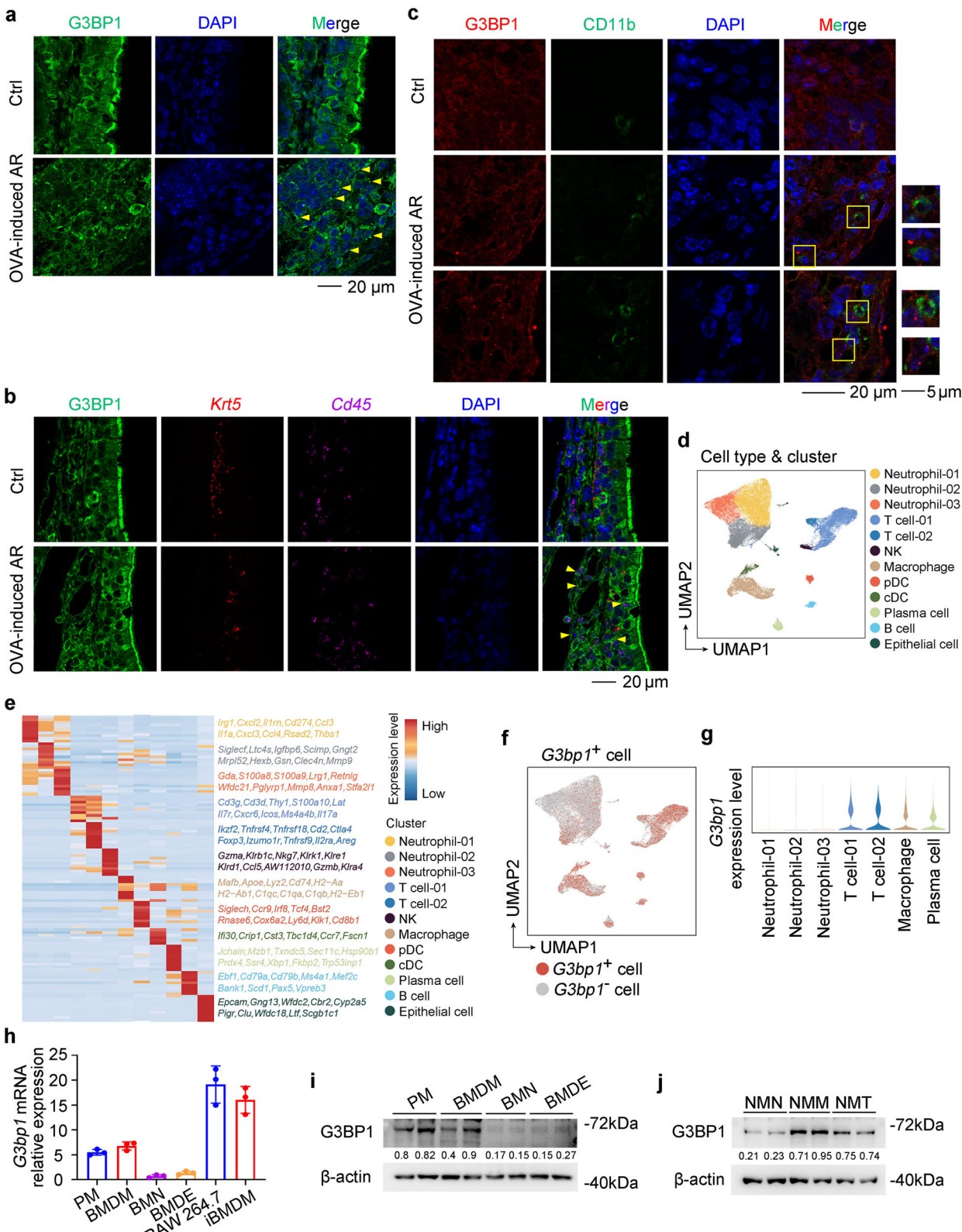

and *G3bp1^mac-/-* mice for the transcriptome analysis and 170 differentially expressed genes were identified (Supplementary Fig. 3a, b), and Kyoto Encyclopedia of Genes and Genomes (KEGG) enrichment and gene set enrichment analysis (GSEA) were performed to characterize the functional groups of these genes. Notably, G3BP1 deficient macrophages upregulated efferocytosis-associated functions, such as endocytosis, protein digestion and absorption, ECM-receptor

interaction and focal adhesion, and showed elevated expression of endocytosis-associated genes (Fig. 3a, b). Thus, we speculated that SG assembly might inhibit the efferocytotic ability of macrophages and impair the clearance of dying immune cells.

We then administered PMs and BMDMs with HDM or NaAsO₂ to induce SG assembly, and co-cultured them with PKH26-labeled apoptotic BMDEs, Jurkat cells, and BMNs, respectively. Flow cytometry and

**Fig. 1 | SGs are assembled in the macrophages of nasal mucosa during AR.**
**a** Immunofluorescence imaging of G3BP1 in the nasal mucosa from control and OVA-induced AR mice. Yellow arrows indicate the assembled SGs.
**b** Immunofluorescence imaging of G3BP1, and RNA-scope staining of *Krt5* and *Cd45* mRNAs in the nasal mucosa from control and OVA-induced AR mice. Yellow arrows indicate the SG and *Cd45* mRNA positive cells. **c** Immunofluorescence imaging of G3BP1 and CD11b in nasal mucosa from control and OVA-induced AR mice. **d** UMAP visualization of CD45⁺ cells in the nasal mucosa of control and OVA-induced AR mice. Each dot corresponds to one single cell colored according to the cell cluster.

**e** Heatmap of gene expression analyzed by scRNA-seq displaying major markers for the main cell types. **f** UMAP visualization of *G3bp1*⁺ cells in the nasal mucosa of control and OVA-induced AR mice. **g** Shown is violin plot of *G3bp1* expression level in neutrophils, T cells, macrophages, and plasma cells. **h** *G3bp1* mRNA expression was detected by qRT-PCR analysis in PMs, BMDMs, BMNs, BMDEs, RAW 264.7 and iBMDM cells (*n* = 3, biological replicates). **i** G3BP1 was detected by Western blot in PMs, BMDMs, BMNs, BMDEs, RAW 264.7 and iBMDM cells. **j** G3BP1 was detected by Western blot in NMMs, NMNs and NMTs. Data are shown as mean ± s.d. or photographs from one representative of three independent experiments.

Incucyte analysis both suggested that SG assembly suppressed the engulfment of apoptotic immune cells by macrophages in vitro (Fig. 3c, d and Supplementary Fig. 3c–e). Interestingly, macrophages with SG assembly were largely defective at engulfing dying cells (Fig. 3e). Similar results were observed through analysis of the peritoneal exudate, which showed that HDM or NaAsO₂ significantly disrupted macrophage efferocytotic ability in vivo (Fig. 3f and Supplementary Fig. 3f–h). Furthermore, in *G3bp1^f/f* and *G3bp1^mac-/-* BMDMs and NMMs, SG assembly-suppressed phagocytosis of PKH26-labeled apoptotic Jurkat and BMDE was blocked by G3BP1 deficiency (Fig. 3g and Supplementary Fig. 3i). The phagocytic ability of PMs and BMDMs under stress conditions was also promoted by *G3bp1* knockdown (Supplementary Fig. 3j, k). Moreover, PKH26-labeled *G3bp1^f/f* macrophages (Red) and PKH67-labeled *G3bp1^mac-/-* macrophages (Green) were co-cultured with CellVue® Claret-labeled apoptotic Jurkat cells (Cyan), and both *G3bp1^f/f* and *G3bp1^mac-/-* macrophages could engulf apoptotic Jurkat cells without stimulus. But under stress conditions, *G3bp1^f/f* macrophages failed to phagocyte apoptotic Jurkat cells while *G3bp1^mac-/-* macrophages still possessed the efferocytotic ability, proving that SG assembly suppresses macrophage efferocytosis (Fig. 3h, i and Supplementary Movie 4). In vivo, *G3bp1^mac-/-* PM engulfed more apoptotic Jurkat and BMDE in mouse peritoneum (Fig. 3j and Supplementary Fig. 3l–p). We also examined *Il-10* mRNA expression in PMs co-cultured with or without apoptotic Jurkat cells, and found more *Il-10* expression in *G3bp1^mac-/-* macrophages than controls (Supplementary Fig. 3q). Thus, these data demonstrate that SG assembly in macrophages impairs efferocytosis.

## SG assembly in macrophages sequesters *Lrp1* mRNA to suppress its translation

To determine the molecular basis by which SG assembly inhibits macrophage efferocytosis, we purified SG core from BMDMs post HDM stimulation and performed the RNA sequencing (RNA-seq) to determine the mRNAs internalized in the SGs of macrophages, as the primary ability of SGs is aggregating RNA-binding proteins and untranslated mRNAs to suppress their translation (Supplementary Fig. 4a–c). 203 transcripts were found to be enriched in the SG cores and termed as SG specific genes (Fig. 4a). Next, we conducted G3BP1 RNA immunoprecipitation sequencing (RIP-seq) in RAW 264.7 macrophages transfected with Flag-tagged G3BP1, and identified 291 increased binding transcripts of G3BP1 post stress with a "GANGANGA" enriched motif (Fig. 4a and Supplementary Fig. 4d, e). When integrating the SG specific genes with the G3BP1 RIP-seq data, three genes, including low-density lipoprotein receptor related protein 1 (*Lrp1*), dynein cytoplasmic 1 heavy chain 1 (*Dync1h1*) and insulin like growth factor 2 receptor (*Igf2r*), were overlapped as the potential targets (Fig. 4a and Supplementary Fig. 4f, g), which were also present in the published SG-enriched-genes dataset obtained from U2OS and NIH3T3 cells[20,21].

LRP1, a multi-ligand receptor, acts as an efferocytosis receptor on macrophages to recognize CALR and PS on apoptotic cells[22,23], and its mRNA was found to be enriched in the SG cores with higher abundance (Supplementary Fig. 4g). Therefore, we focused on *Lrp1*, and presumed that it was the target for SG assembly-impaired macrophage

efferocytosis. The association between G3BP1 and *Lrp1* mRNA was determined by RIP-qRT-PCR in both BMDMs and PMs, which could be enhanced under HDM (Fig. 4b), and the nuclear transporter factor 2 (NTF2) domain of G3BP1 was responsible for their binding, which is well-conserved to its human homolog (Fig. 4c and Supplementary Fig. 4h). Furthermore, we conducted rescue experiment in *G3bp1^mac-/-* macrophages, and confirmed that the NTF2 domain of G3BP1 was responsible for its binding with *Lrp1* mRNA and the inhibition of macrophage engulfment (Supplementary Fig. 4i, j). We also performed RNA-scope, FISH and RNA pull-down assays to confirm that G3BP1 directly associated with *Lrp1* mRNA especially in SGs under stress (Fig. 4d, e and Supplementary Fig. 4k). As recent studies have reported that G3BP1, as the central hub of SG, modulates the stability and translation of mRNAs[2,24], we performed immunoblotting and flow cytometry, and found that LRP1 protein level in macrophages was significantly reduced post HDM or NaAsO₂ administration, while *Lrp1* mRNA level was not altered (Fig. 4f, g and Supplementary Fig. 4l, m). Furthermore, macrophages from *G3bp1^mac-/-* mice exhibited increased LRP1 protein level while unaltered mRNA level as compared to *G3bp1^f/f*, which was also confirmed by *G3bp1* knockdown (Fig. 4h–j and Supplementary Fig. 4n–p). Thus, cytoplasmic SG assembly decreases the protein level of LRP1, the membrane efferocytosis receptor, to impair macrophage efferocytosis, thereby exacerbating AR progression.

## Internal m⁷G-modified *Lrp1* mRNA is shuttled by QKI7 into SGs, thus repressing its translation

To assess whether SG assembly-decreased LRP1 protein expression relied on translational arrest or protein degradation, as its mRNA level was unaltered, we performed polysome profiling and protein degradation assays in macrophages. SG assembly under HDM stimulus significantly repressed the translation of *Lrp1* mRNA, while it was promoted by G3BP1 deficiency (Fig. 5a, b and Supplementary Fig. 5a, b). However, the inhibition of neither proteasome nor lysosome in macrophages could significantly increase LRP1 protein expression (Supplementary Fig. 5c–f). Evidence is emerging that, under stress condition, internal *N⁷*-methylguanosine (m⁷G)-modified mRNAs are believed to be shuttled into the SGs for modulating their translational efficiency or mRNA stability[24]. Concomitantly, we found abundant internal m⁷G-modification within *Lrp1* mRNA in macrophages through mRNA m⁷G-methylated RIP-seq (m⁷G-MeRIP-seq) and m⁷G-MeRIP-qRT-PCR, with similar sequence characteristic for its binding with G3BP1 elucidated by RIP-seq (Fig. 5c, d). We also re-analyzed previously published m⁷G-MeRIP-seq data[24,25], and found that *Lrp1* mRNA was also m⁷G-modified in MEF and some other cell lines, thus suggesting that the m⁷G-modification of *Lrp1* mRNA may be universal in cells (Supplementary Fig. 5g). Additionally, G3BP1 deficiency did not affect the internal m⁷G-modification of *Lrp1* mRNA (Supplementary Fig. 5h). Thus, *Lrp1* mRNA is m⁷G-modified, which may be shuttled into the assembled SGs to suppress its translation in macrophages post stress.

It has been reported that Quaking protein 7 (QKI7), a newly found internal m⁷G reader, fine-tunes the stability and translation of a set of internal m⁷G-modified transcripts under stress by shuttling them into SGs[24]. Therefore, we presumed that internal

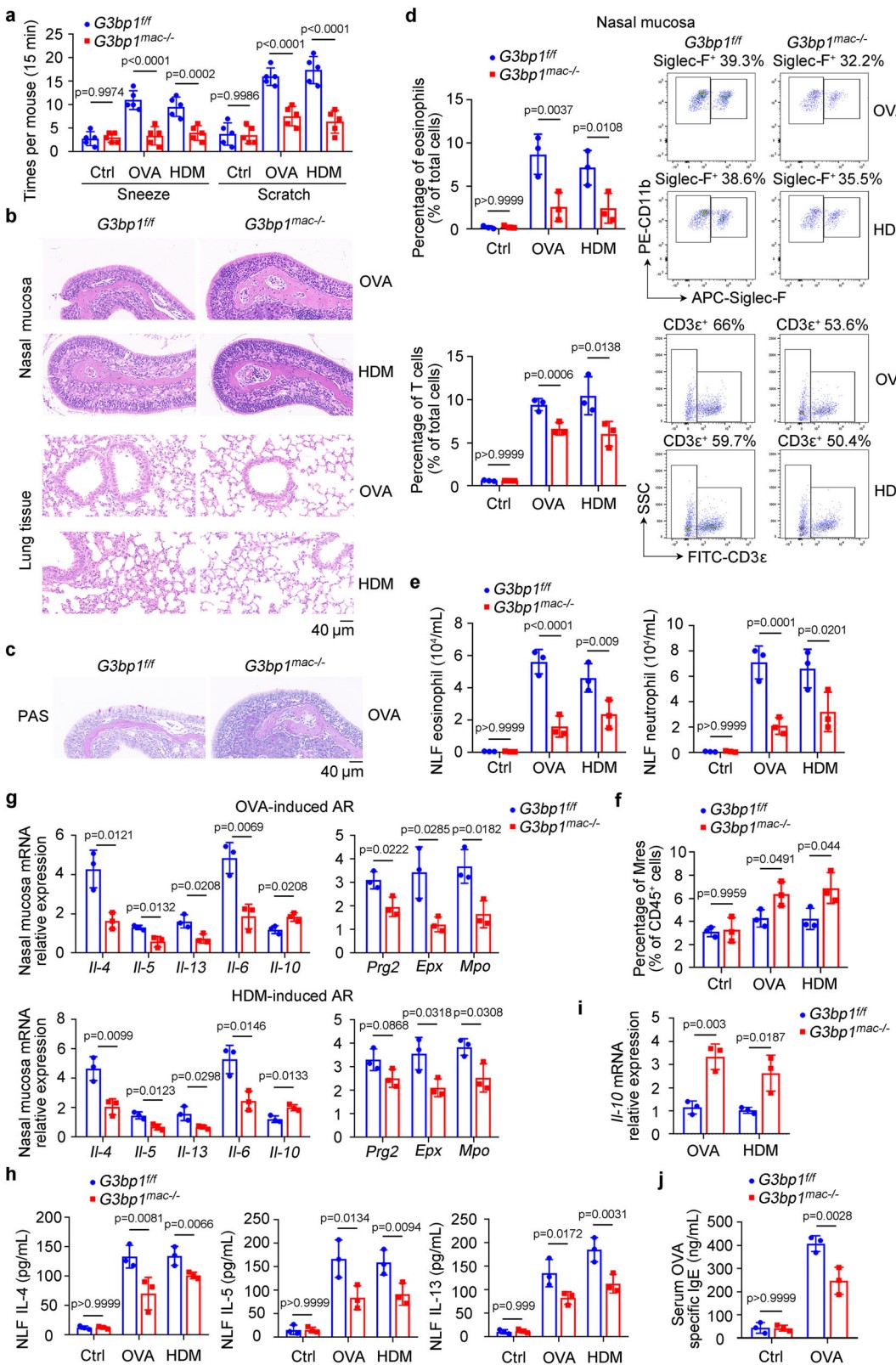

m[7]G-modified *Lrp1* mRNA was recognized by QKI7 and then shuttled accordingly into the assembled SGs of macrophages under stress. The co-localization between QKI7 and G3BP1 was found in the SGs of macrophages under HDM stress, which recruited m[7]G-modified *Lrp1* mRNA into the assembled SGs (Fig. 5e, f and Supplementary Fig. 5i). Moreover, the association between QKI7 and *Lrp1* mRNA was determined by RIP-qRT-PCR

assay in macrophages, which was confirmed by the re-analysis of the published RIP-seq dataset from HepG2 and U2OS cells (Fig. 5g and Supplementary Fig. 5j)[24]. Additionally, K homology (KH) domain of QKI7, a well-conserved domain, was responsible for their binding (Fig. 5h and Supplementary Fig. 5k). Furthermore, we performed RNA pull-down assay to confirm that QKI7 directly associated with *Lrp1* mRNA (Fig. 5i). Additionally, the association

**Fig. 2 | SG assembly in macrophages promotes AR progression.** *G3bp1^f/f* and *G3bp1^mac-/-* mice were administrated with OVA or HDM to induce AR symptoms as depicted in Supplementary Fig. 2e. **a** Times of sneezes and scratches in each mouse were counted in 15 min after last i.n. challenge from control, OVA and HDM group (*n* = 5, biological replicates). (Tukey's HSD). **b**, **c** H&E and PAS staining of the nasal mucosa and lung tissues from AR mice. **d** Flow cytometry of eosinophils and T cells in the nasal mucosa of AR mice (*n* = 3, biological replicates). (Tukey's HSD). **e** Flow cytometry of eosinophils and neutrophils in the NLF obtained from control and AR mice (*n* = 3, biological replicates). (Tukey's HSD). **f** Flow cytometry of Mres in the nasal mucosa of control and AR mice (*n* = 3, biological replicates). (Tukey's HSD).

**g** mRNA expressions of indicated cytokines, *Prg2* (proteoglycan 2, pro-eosinophil major basic protein), *Epx* (eosinophil peroxidase), and *Mpo* (myeloperoxidase) were detected by qRT-PCR in the nasal mucosa of AR mice (*n* = 3, biological replicates). (two-tailed unpaired Student's *t*-test). **h** ELISA assay of IL-4, IL-5, and IL-13 in the NLF obtained from control and AR mice (*n* = 3, biological replicates). (Tukey's HSD). **i** mRNA expressions of *Il-10* were detected by qRT-PCR in NMMs of AR mice (*n* = 3, biological replicates). (Tukey's HSD). **j** Serum OVA-specific IgE level in the control and AR mice (*n* = 3, biological replicates). (Tukey's HSD). Data are shown as mean ± s.d. or photographs from one representative of three independent experiments.

---

between QKI7 and *Lrp1* mRNA was not influenced by G3BP1 (Supplementary Fig. 5l), and the m^7^G-modification of *Lrp1* mRNA was not influenced by QKI7 (Supplementary Fig. 5m, n). Given that SG assembly may suppress the translation of the internalized m^7^G modified mRNAs, we then sought to assess whether QKI7, as a m^7^G reader, could modulate the fate of *Lrp1* mRNAs in SGs. We re-analyzed the previously published ribosome sequencing (Riboseq) data, and found that overexpression of QKI7 significantly attenuated the translation efficiency of *LRP1* mRNA under stress conditions, which was even more prominent than *GSK3B* mRNA, a translational arrest target described in that published study (Supplementary Fig. 5o)[24]. Therefore, we performed a polysome profiling assay under HDM stimulus, and QKI7 overexpression significantly reduced the enrichment of *Lrp1* mRNA in polysome fractions under stress (Fig. 5j and Supplementary Fig. 5p), confirming that QKI7 sequesters *Lrp1* mRNA within the assembled SGs and suppresses its translation under stress. Thus, the internal m^7^G-modified *Lrp1* mRNA is shuttled by the m^7^G reader QKI7 into the assembled SGs of macrophages under stress to attenuate its translational efficiency, which may then impair macrophage efferocytosis to promote AR progression.

### The SG-promoted AR depends on the inhibition of efferocytosis receptor LRP1

To directly address the role of the assembled SG-inhibited LRP1 expression in suppressing macrophage efferocytosis and promoting AR, we generated macrophage-specific *Lrp1* knockout (*Lrp1^mac-/-*) mice by crossing *Lrp1* floxed (*Lrp1^f/f*) to *Csf1r-Cre* mice (Supplementary Fig. 6a, b). In the OVA-induced AR mouse model, opposite to *G3bp1^mac-/-* mice, *Lrp1^mac-/-* mice displayed more severe AR symptoms with increased sneezing and scratching counts (Fig. 6a), aggravated mucosa damage with higher AR histological score (Fig. 6b and Supplementary Fig. 6c, d), elevated infiltration of eosinophils, neutrophils and T cells (Fig. 6c), increased cytokines expression in the nasal mucosa (Supplementary Fig. 6e), and higher cytokines production in NLF (Fig. 6d) than those of *Lrp1^f/f* mice. Additionally, serum OVA-specific IgE concentration was also elevated in *Lrp1^mac-/-* mice (Supplementary Fig. 6f). We also measured *Il-4* mRNA expression during Th2 differentiation, and found no significant difference between *Lrp1^f/f* and *Lrp1^mac-/-* (Supplementary Fig. 6g). Next, both in vivo and in vitro efferocytosis assay determined that LRP1 deficiency disturbed macrophage phagocytotic ability to clear apoptotic cells, thus promoting AR progression (Fig. 6e–g). Furthermore, recombinant mouse LRPAP protein, a LRP1 blocker, was utilized in the efferocytosis assay, confirming the inhibition of macrophage efferocytosis and reduction of *Il-10* expression by the suppression of LRP1 (Supplementary Fig. 6h, i).

Furthermore, we crossed and generated macrophage-specific *G3bp1* and *Lrp1* double knockout (*G3bp1^mac-/-Lrp1^mac-/-*) mice by crossing *G3bp1^f/f Lrp1^f/f* to *Csf1r-Cre* mice to examine whether the assembled SG-promoted AR was dependent on the suppressed downstream LRP1 (Supplementary Fig. 6b). *G3bp1^mac-/-* under *Lrp1^mac-/-* failed to alleviate the OVA-induced AR symptoms, and the symptoms were similar between *G3bp1^mac-/-Lrp1^mac-/-* and *Lrp1^mac-/-* mice (Fig. 6h and Supplementary Fig. 6j),

which was confirmed by the analysis of mucosa histological score (Fig. 6i and Supplementary Fig. 6k), immune cells infiltration (Fig. 6j), cytokines production (Supplementary Fig. 6l), and serum OVA-specific IgE (Supplementary Fig. 6m). The decreased macrophage efferocytosis by *Lrp1^mac-/-* was also not alleviated by *G3bp1^mac-/-* in vivo (Fig. 6k). Hence, G3BP1 deficiency failed to promote macrophage efferocytosis and suppress AR development under LRP1 deficient background, suggesting that the assembled SG-impaired macrophage efferocytosis is dependent on the suppressed downstream LRP1.

### Preventing SG assembly with G3BP1-specific inhibitor alleviates AR progression

We found above that G3BP1 associated with *Lrp1* mRNA through its NTF2 domain (Fig. 4c), and noticed that a recently reported G3BP1 inhibitor (G3Ia, FAZ-3532), also targeting the NTF2 domain, disrupted the co-condensation of G3BP1 and RNAs, and inhibited SG formation[26]. Thus, we treated BMDMs with G3Ia, followed by the administration of HDM, and SG assembly was significantly inhibited in a dose-dependent manner (Supplementary Fig. 7a). To identify the effect of G3Ia on AR progression, OVA-induced AR mice were intranasal (i.n.) treated with vehicle, NaAsO$_2$ (0.015 mg/kg), or G3Ia (0.05 mg/kg) once a day, together with OVA challenge for 7 days, or HDM-induced AR mice were i.n. treated with vehicle or G3Ia together with HDM challenge for 5 days as presented (Supplementary Fig. 7b). Group of NaAsO$_2$, the SG inducer, exhibited more severe AR symptoms, while G3Ia treatment effectively alleviated AR progression (Fig. 7a). Simultaneously, the treatment using G3Ia significantly improved mucosa integrity, inhibited immune cells infiltration, cytokines production, and serum OVA-specific IgE (Fig. 7b–f and Supplementary Fig. 7c, d). The G3Ia treatment also significantly alleviated HDM-induced AR symptoms and mucosa damage (Supplementary Fig. 7e, f). Additionally, G3Ia treatment successfully rescued the defect of macrophage efferocytosis under stress and restored LRP1 expression at the protein level (Supplementary Fig. 7g, h). Hence, the prevention of SG assembly using G3BP1-NTF2 domain-specific inhibitor alleviates AR symptoms, which may be effective in the targeted therapy.

### SGs are assembled in the macrophages of nasal mucosa from AR patients

We then examined the human nasal mucosa tissues from patients, which were collected during nasal surgery, including deviated nasal septum without AR (as healthy controls), and AR patients (Supplementary Fig. 7i). The SG formation in the nasal mucosa from healthy controls and AR patients was analyzed, and we found that SGs were assembled in the macrophages of nasal mucosa from AR patients, but not in the healthy controls, thereby suggesting that the assembled SGs in nasal mucosa macrophages may participate in the etiology of human AR (Fig. 7g and Supplementary Fig. 7j, k). Altogether, we discover that SG assembly under stress sequestered the internal m^7^G-modified *Lrp1* mRNA, recognized and shuttled by the m^7^G reader QKI7, and then suppressed the translation of efferocytosis receptor *Lrp1* mRNA, thus impairing macrophage efferocytosis of apoptotic immune cells to aggravate AR progression (Fig. 7h).

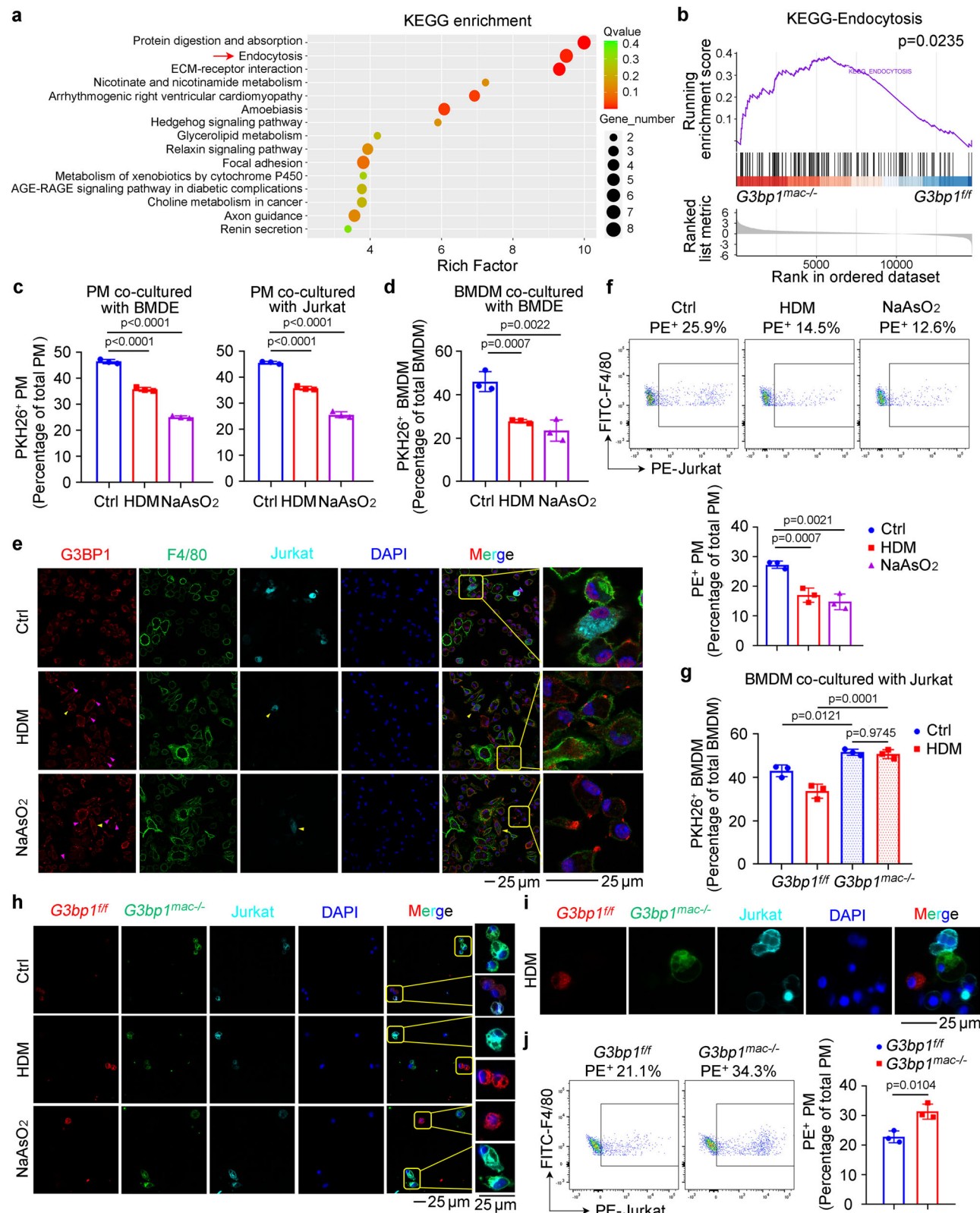

## Discussion

In the past decade, the critical roles of macrophages in the innate immunity, infectious diseases, and tumor microenvironment have attracted much attention and been thoroughly demonstrated, especially their efferocytotic ability to clear apoptotic immune cells, resolve inflammation, and maintain tissue homeostasis[15,27]. However, the function of macrophage efferocytosis in allergic diseases especially AR is still elusive. Here, we identify that during AR and upon stimulus, SGs are assembled specifically in the macrophages within nasal mucosa, which then impairs the efferocytotic ability to clear apoptotic immune cells and increase the production of type II cytokines, thus aggravating AR progression. Hence, the roles of SG assembly and macrophage efferocytosis in AR development have been exhibited, and the potential therapeutic target has been suggested in this study.

**Fig. 3 | SG assembly in macrophages impairs efferocytosis. a**, Pathway enrichment analysis of the differentially expressed genes according to KEGG database in macrophages from *G3bp1^{f/f}* and *G3bp1^{mac-/-}* mice. **b** The enrichment plot of the GSEA analysis for RNA-seq data on endocytosis-related gene sets (KEGG). **c, d** HDM (100 μg/well) or NaAsO$_2$ (50 μM)-inhibited in vitro macrophage efferocytosis of apoptotic cells (labeled with PKH26) was measured by flow cytometry (*n* = 3, biological replicates). (Dunnett's multiple comparisons test). **e** Representative immunofluorescence assay showing efferocytosis of apoptotic cells (labeled with Claret) by macrophages (labeled with F4/80). Yellow arrows indicate phagocytotic macrophages. Magenta arrows indicate macrophages with SG assembly. **f** HDM (100 μg/well) or NaAsO$_2$ (50 μM)-inhibited in vivo efferocytosis of apoptotic Jurkat cells (labeled with PKH26) by PMs was measured by Flow cytometry (*n* = 3, biological replicates). (Dunnett's multiple comparisons test). **g** In vitro efferocytosis assay of

PKH26-labeled apoptotic Jurkat cells engulfed by *G3bp1^{f/f}* and *G3bp1^{mac-/-}* macrophages was measured by flow cytometry (*n* = 3, biological replicates). (Tukey's HSD). **h** Representative immunofluorescence assay showing efferocytosis of apoptotic cells (labeled with Claret) by macrophages (*G3bp1^{f/f}* macrophages labeled with PKH26 red and *G3bp1^{mac-/-}* macrophages labeled with PKH67 green). **i**, Living cell imaging showing efferocytosis of apoptotic cells (labeled with Claret) by macrophages (*G3bp1^{f/f}* macrophages labeled with PKH26 red and *G3bp1^{mac-/-}* macrophages labeled with PKH67 green). **j** In vivo efferocytosis assay of PKH26-labeled apoptotic Jurkat cells engulfed by *G3bp1^{f/f}* and *G3bp1^{mac-/-}* macrophages was measured by flow cytometry (*n* = 3, biological replicates). (two-tailed unpaired Student's *t*-test). Data are shown as mean ± s.d. or photographs from one representative of three independent experiments.

The roles of epithelial cells, Th2 cells, B cells, ILC2s, and their crosstalk in the regulation of the innate and adaptive immune responses in the pathogenesis of allergic airway diseases have been intensively investigated[7,28]. For the epithelial cells, it was reported that SG assembly in bronchial epithelial cells facilitated the nuclear-cytoplasmic transport of IL-33, the important epithelial cell-derived alarmin, to promote its production and aggravate allergic bronchopulmonary airway inflammation[29]. However, we did not find the assembled SGs in the epithelial cells of nose, which may be due to the different origins, differentiation processes, structures, and biological functions between nasal and bronchial epithelial cells. Moreover, our scRNA-seq data of CD45$^+$ cells from the nasal mucosa of AR mice identified that *G3bp1* was also expressed in T and B cells, both were critical cells in the pathogenesis of AR. Although we did not find the significant SG assembly in these two cell types, it remains the important scientific questions that which stimulus can induce the assembly of SGs in these cells and what are the functions of SG formation in their pro-allergic property, especially that whether SGs are assembled in these cells during the early sensitization period is still not investigated.

Macrophage efferocytosis clears apoptotic cells to prevent secondary necrosis and the release of proinflammatory necrotic cell debris, and the defects of apoptotic cell uptake contribute to the uncontrolled inflammation and impaired tissue homeostasis[6,15]. Previous studies demonstrated that defective phagocytosis of apoptotic eosinophils and neutrophils by depressed efferocytosis of airway macrophages delayed the resolution of inflammation and contributed to the pathogenesis of asthma[30,31]. Together with our study that the impaired macrophage efferocytosis participates in AR development, the design of new approaches to enhance macrophage efferocytic ability may have considerable potential in the treatment of allergic airway diseases. The reduced efferocytosis receptor LRP1 by SG assembly is determined to be responsible for the impaired efferocytosis of macrophages upon stimulus. It raises an interesting question that whether the defective efferocytotic ability of stressed macrophages is a common mechanism in the pathogenesis of series of inflammatory or allergic diseases, which needs to be investigated extensively. Additionally, the potential roles of other important efferocytosis regulators, such as the 'don't eat me' signal CD47, are also needed to be assessed in the pathogenesis of AR.

In the macrophages without stimulus, we find that G3BP1 and QKI7 distribute evenly in the cytoplasm. When SGs are assembled, the internal m$^7$G-modified *Lrp1* mRNA is shuttled by QKI7 into the SG, which is consistent with the determined direct association between G3BP1 and QKI7[24]. It would be interesting and necessary to investigate the underlying mechanism responsible for this induced association between G3BP1 and QKI7. A previous report indicates that the conformational switch and clustering of G3BP drive SG assembly upon stress[32], whether this also facilitates the association between G3BP1 and QKI7 needs to be illustrated. Additionally, as stress induces various kinds of post-translational modifications of G3BP1, including ubiquitination, methylation, and acetylation[33-35], whether these modifications

can result in the induced association between G3BP1 and QKI7 is still unknown. Since G3BP1 is the central node of protein-RNA interaction network in SGs[36], QKI7, with the ability to associate with G3BP1, may be one of the inherent components in the assembled SGs. A current study shows that RNA damage can induce the formation of specific DHX9 SGs, which are different from the classic SGs and share different core proteins and RNA profiling[37]. It would be interesting and necessary to investigate whether QKI7 is also included in the DHX9 SGs or specifically in the formation of traditional SGs. Moreover, we present that SG assembly in macrophages represses efferocytosis in AR pathogenesis, which is different from the role of SGs in bronchial epithelial cells to promote IL-33. This discrepancy also suggests the different biological functions of SGs in different types of cells[29], which is closely related to the properties and characteristics of the cell types.

In the macrophages of nasal mucosa, we identify that internal m$^7$G-modified *Lrp1* mRNA is shuttled by QKI7 into the assembled SGs to mediate its translational repression, which impairs macrophage efferocytosis to aggravate AR. As reported, methyltransferase like 1 (METTL1)-WD repeat domain 4 (WDR4) complex is demonstrated to methylate a subset of internal m$^7$G sites within mRNAs, and the depletion of them not only attenuates the enrichment of internal m$^7$G-modified mRNAs in the assembled SGs, but also impairs the binding of QKI7 with the m$^7$G-modified targets[24]. Thereafter, it would be intriguing to investigate whether METTL1 or WDR4 also participate in the regulation of macrophage efferocytosis, pathogenesis of AR, tissue repair and healing, resolution of inflammation, and the development of associated inflammatory diseases.

As reported by several studies, a set of compounds were designed to disrupt the formation of SGs, including glutamate dehydrogenase1/2 (GLUD1/2) inhibitor (Epigallocatechin Gallatein), c-Jun N-terminal kinase (JNK) activator (anisomycin), and eukaryotic initiation factor 2α (eIF2α) activator (ISRIB, a brain-penetrant inhibitor of integrated stress response)[38-41]. However, all these compounds are not designed to directly target G3BP1. Compared with the previous inhibitors, we introduced a currently reported small molecule compound G3Ia, specifically targeting the NTF2 domain of G3BP1, which was determined to be responsible for the association of G3BP1 with *Lrp1* mRNA in this study. As expected, the intranasal treatment of G3Ia successfully inhibited SG assembly, promoted macrophage efferocytosis, and alleviated the pathogenesis of AR, suggesting the considerable therapeutic potential of targeting G3BP1 SG assembly using this inhibitor intranasally for the treatment of AR. At present, nasal spray of glucocorticoid or oral administration of histamine antagonists are the primary treatments for AR patients, but the low persistence of curative effect and concern for side effects of long-term use are still the main problems[7]. Recent specific immunotherapy is also characterized by lengthy duration (2–3 years), complex procedures, and low patient adherence, with unsatisfactory long-term effects presented by the trials[42]. As our study has presented the assembled SGs in nasal mucosa macrophages of AR patients, the nasal spray of G3Ia may be promising in

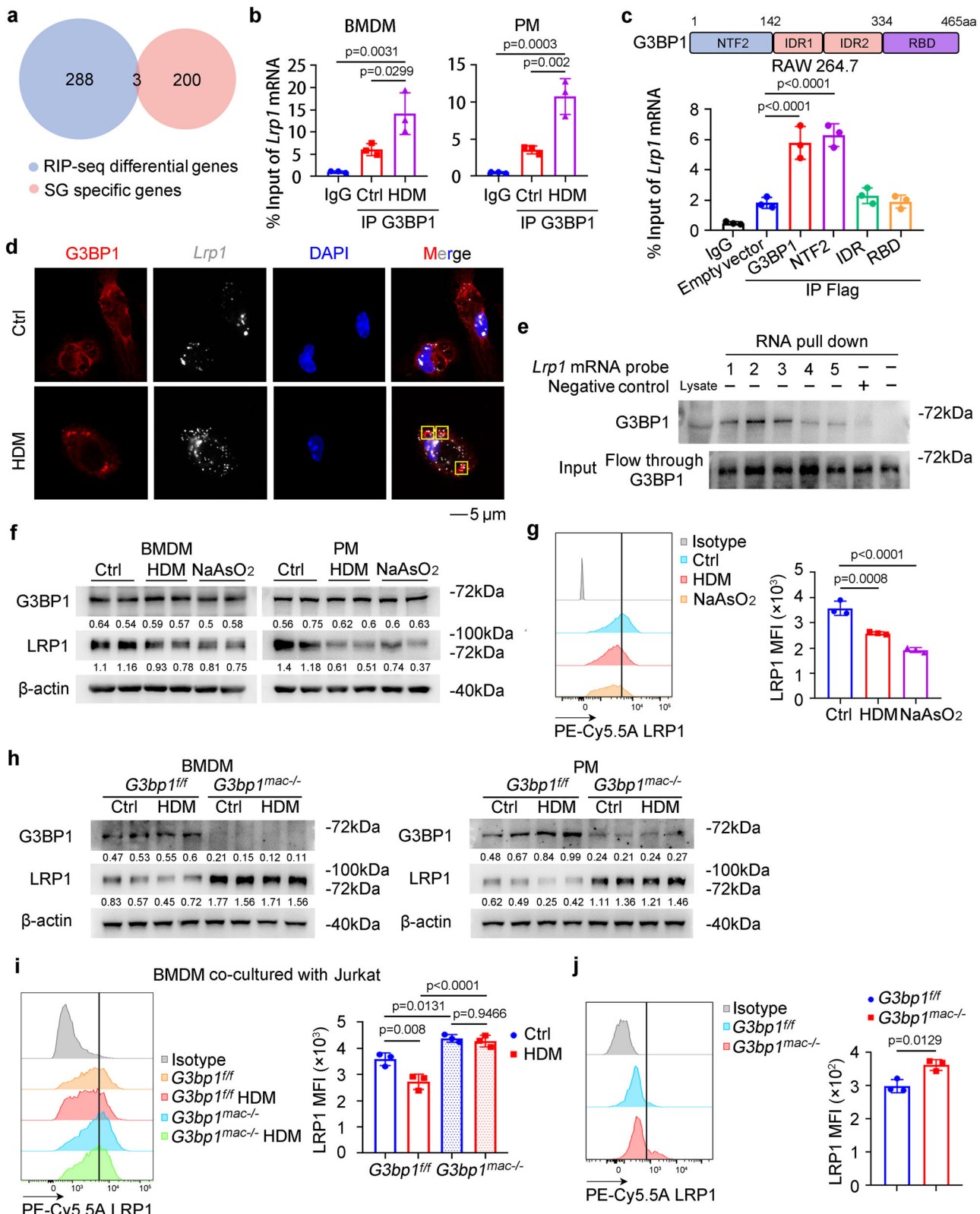

the treatment of AR patients, which is needed to be determined in the clinical investigation.

## Methods

### Tissues
Human nasal mucosa samples were obtained from patients during nasal surgery in Second Military Medical University (Shanghai, China), including deviated nasal septum without AR (as healthy controls), and AR patients with the diagnosis based on the symptoms and allergen-specific IgE level class ≥2. All the tissue samples in this study were collected with written informed consent from the patients, and the experiments were approved by the Institute Research Ethics Committee of Second Military Medical University (2024SL128).

**Fig. 4 | SG assembly in macrophages sequesters *Lrp1* mRNA to suppress its translation. a** Venn diagram indicating overlap of mRNAs with SG-enrichment and increased G3BP1-binding under stress conditions (using the OmicStudio tools at https://www.omicstudio.cn/tool). **b** The association between G3BP1 and *Lrp1* mRNA in primary macrophages was determined by RIP-qRT-PCR ($n = 3$, biological replicates). (Tukey's HSD). **c** The association between the NTF2 domain of G3BP1 and *Lrp1* mRNA in RAW 264.7 cells was determined by RIP-qRT-PCR ($n = 3$, biological replicates). (Tukey's HSD). **d** Immunofluorescence of G3BP1 and RNA-scope imaging of *Lrp1* mRNA in PMs. **e** The association between G3BP1 and *Lrp1* mRNA in PMs was determined by RNA pull-down assay. **f** G3BP1 and LRP1 protein levels in primary macrophages stimulated with HDM (100 μg/well) or NaAsO$_2$ (50 μM) were

examined by Western blot. **g** LRP1 protein level on the surface of PMs stimulated with HDM (100 μg/well) or NaAsO$_2$ (50 μM) were examined by flow cytometry ($n = 3$, biological replicates). (Dunnett's multiple comparisons test). **h** G3BP1 and LRP1 protein levels in primary macrophages of *G3bp1$^{f/f}$* and *G3bp1$^{mac-/-}$* mice stimulated with HDM (100 μg/well) were examined by Western blot. **i** LRP1 protein level on the surface of BMDMs of *G3bp1$^{f/f}$* and *G3bp1$^{mac-/-}$* mice stimulated with HDM (100 μg/well) were examined by flow cytometry ($n = 3$, biological replicates). (Tukey's HSD). **j** LRP1 protein level on the surface of PMs of *G3bp1$^{f/f}$* and *G3bp1$^{mac-/-}$* mice were examined by flow cytometry ($n = 3$, biological replicates). (two-tailed unpaired Student's *t*-test). Data are shown as mean ± s.d. or photographs from one representative of three independent experiments.

## Mice

C57BL/6 J mice (Female; 6–8 weeks old) were obtained from the Joint Ventures Sipper BK Experimental Animal Company (Shanghai, China). *G3bp1$^{f/f}$* (No. S-CKO-09896) mice were constructed by Cyagen Biosciences Corporation (Suzhou, China). *Ly6g*-Cre (NM-KI-200219) mice and *Siglecf*-Cre (NM-KI-231128) mice were constructed by Shanghai Biomodel Organism Science & Technology Development Corporation (Shanghai, China). *Lrp1$^{f/f}$* mice (No. T064127) and *Csf1r*-Cre mice (No. T005640) were constructed by GemPharmatech (Nanjing, China). *Lyz2*-Cre mice (No. 004781) were obtained from The Jackson Laboratory. Genotyping was done by PCR analysis on genomic DNA extracted from mice tails as previously described[43]. All animal experiments were undertaken in accordance with the National Institute of Health Guide for the Care and Use of Laboratory Animals, with the approval of the Scientific Investigation Board of Second Military Medical University, Shanghai, China. All mice were bred and housed in a specific pathogen-free (SPF) animal facility at a temperature of $23 \pm 2\,°C$, relative humidity range between 40-70% environment with a 12 h light/dark cycle. The experimental and control mice were co-housed.

## Primer sequences for mice identification

*G3bp1$^{f/f}$* forward: 5′- CAG TCT GCA TGT CTT TAA CCT CTT −3′, reverse: 5′- ACA CAA GAG TGG TGG ATA GTA TCA G −3′; *Lrp1$^{f/f}$* forward: 5′- TAC AAG TGC GTG CTA CTG CAT ACA G-3′, reverse: 5′- GTA GAG ACC TAA GGC ATC AGG TCC C −3′; *Lyz2*-Cre forward: 5′- CCC AGA AAT GCC AGA TTA CG −3′, reverse: 5′- CTT GGG CTG CCA GAA TTT CTC −3′; *Ly6g*-Cre forward: 5′-CTG CAA CCT GGT CAG AGA GG-3′, reverse: 5′- AGC ATT GGA GTC AGA AGG GC-3′; *Siglecf*-Cre forward: 5′-CAG GCA ACA GAC TCA GGG AG-3′, reverse: 5′-CCT GTT GTT CAG CTT GCA CC-3′; *Csf1r*-Cre forward: 5′-GGA CTA TGC TAA CCT GCC AAG C-3′, reverse: 5′-GGA ATG CTC GTC AAG AAG ACA G-3′.

## Reagents

Antibodies specific to G3BP1 (45656), LRP1 (64099), eIF4G (2498), horseradish peroxidase coupled secondary antibodies (7074 and 7076) and normal rabbit IgG (2729) were purchased from Cell Signaling Technology (Danvers, MA). Polyclonal antibody specific to G3BP1 (13057-2-AP), polyclonal antibody specific to eIF3b (10319-1-AP) and monoclonal antibody specific to G3BP1 (66486-1-Ig) were purchased from Proteintech (Wuhan, China). Antibodies specific to CD45 (GB113886), CD3 (GB11014), CD11b (GB115689), CD19 (GB11061-1), and β-actin (12044) were purchased from Servicebio (Wuhan, China). Antibody specific to Flag-tag (F1804) were from Sigma-Aldrich (St. Louis, MO). Antibody specific to m$^7$G (RN017M) was from MBL International Corporation (Schaumburg, IL). Antibody specific to QKI7 (N183/15) was from NIH NeuroMab Facility Corporation (Davis, CA). Alexa Fluor 555 coupled secondary antibodies (A-21428 and A-21422), Alexa Fluor 488 coupled secondary antibodies (A-11008 and A-11001), Alexa Fluor 647 coupled secondary antibody (A-21244), and antibody specific to CD91-NovaFluor Yellow 730 (H073T03Y07-A) were purchased from Invitrogen (Carlsbad, CA). Antibodies specific to CD45-BV711 (103147), CD11b-PE (101207), Ly6G-Percp.cy5.5 (127615), SiglecF-

APC (155507), F4/80-FITC (157310), CD19-APC-CY7 (152411), CD3ε-FITC (100305) and TruStain FcX CD16/32 (101320) were from BioLegend (San Diego, CA). Protease inhibitor cocktail (11836170001), Ovalbumin (A5503), CellVue Claret Far Red Fluorescent Cell Linker Midi Kit (MINCLARET), PKH26 Red Fluorescent Cell Linker Midi Kit (MIDI26), PKH67 Red Fluorescent Cell Linker Midi Kit (MIDI67), sodium arsenite (S7400), hyaluronidase (V900833), DNase I (10104159001), Magna RIP RNA binding protein immunoprecipitation (RIP) Kit (17-700) and Magna ChIP Protein A + G Magnetic Beads (16-663) were purchased from Sigma Aldrich. Anti-FLAG M2 Magnetic Beads (M8823), Pure-Proteome Protein A/G Mix Magnetic Beads (LSKMAGAG), and sucrose (573113) were purchased from Millipore. BD Horizon Fixable Viability Stain 510 (564406), Fixation and Permeabilization Solution (554722), and Perm/Wash Buffer (554723) were purchased from BD Biosciences. Dynabeads mRNA Purification Kit (61006), Sodium Pyruvate (11360070), GlutaMax Supplement (35050061), β-mercaptoethanol (21985023), Imject Alum (77161), Pierce ECL Western (PI32106), SUPERase, In RNase Inhibitor (AM2694), Pierce Protein A/G Magnetic Beads (88803), and Pierce Magnetic RNA-Protein Pull-Down Kit (20164) were purchased from Thermo Fisher Scientific (Waltham, MA). RNA Clean & Concentrator-5 (R1015) was purchased from Zymo Research (Orange County, CA). HDM (XPB91D3A2.5) was purchased from Stallergenes Greer. Cycloheximide (HY-12320), MG132 (HY-13259), and G3Ia (HY-162288) were purchased from MedChemExpress. Recombinant mouse LRPAP protein (4480-LR) was purchased from R&D system. Transfection reagent jetPRIME (101000001) was purchased from Polyplus (Illkirch, France). Transfection reagent Lipofectamin RNAiMAX (13778150) and TRIzol (15596026CN) were purchased from Invitrogen. FuGENE HD Transfection Reagent (HD-1000) was purchased from Fugene LCC (Middleton, Wisconsin). EasySep Mouse Neutrophil Enrichment Kit (19762), EasySep Mouse F4/80 Enrichment Kit (100-0659) and EasySep Mouse T Cell Isolation Kit (9851RF) were purchased from Stem Cell Technologies (Vancouver, BC). Recombinant mouse IL-4 (574304) was purchased from Biolegend. Recombinant murine IL-5 (215-15), murine stem cell factor (SCF) (250-03), and murine M-CSF (315-02) were purchased from PeproTech (Cranbury, NJ). Recombinant murine Flt3 ligand (ab270071) and Mounting Medium with DAPI (ab104139) were purchased from Abcam (Cambridge, UK). NEB Next Magnesium RNA Fragmentation Module (E6150S) and T4 polynucleotide kinase (M0201S) were purchased from New England Biolabs (Ipswich, MA). Tobacco Decapping Plus 2 (94) was purchased from Enzymax (Lexington, KY). DMEM (11965092), Fetal Bovine Serum (FBS, 10099141 C), RPMI 1640 (11875093), Penicillin Streptomycin (15140122), HEPES (15620080), Non-essential Amino Acids solution (100×, 11140076), Collagenase type I (17100017), and Trypsin-EDTA (0.05%) (25300054) were from Gibco (Shanghai, China). LEGEND MAXTM Mouse IL-4 ELISA Kit (431107) was purchased from Biolegend. Mouse IL-13 Quantikine ELISA Kit (M1300CB) and mouse IL-5 Quantikine ELISA Kit (M5000) were purchased from R&D Systems (Minneapolis, MN). Mouse OVA-specific IgE ELISA Kit (BPE20761) was purchased from Lengton Biology (Shanghai, China).

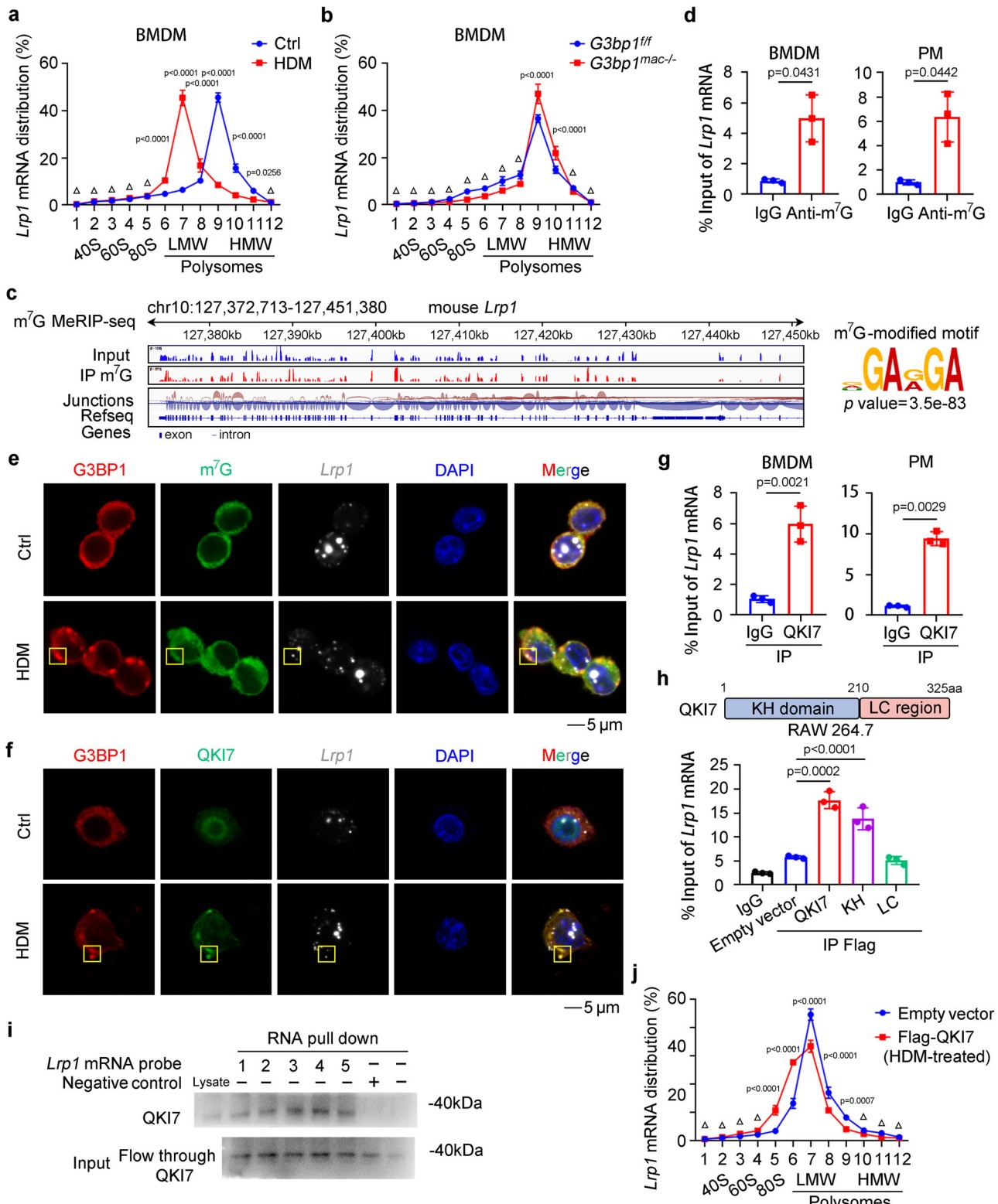

## AR mouse models

Mice were sensitized on days 0, 7, and 14 by intraperitoneal (i.p.) injections of OVA (0.1 mg) together with Imject Alum (2 mg) in 200 μL PBS. From days 21 to 27, these mice were intranasally (i.n.) challenged with 2 mg of OVA in 20 μL PBS without anesthesia. On day 27, the AR symptoms of mice were measured. In detail, after the last nasal drip, each mouse was placed alone in the cage position for 15 min, and the number of scratching and sneezing of each mouse within 15 min was counted. On day 28, these mice were anesthetized for further analysis.

HDM-induced AR mouse model was constructed according to the previously reported protocols[44]. Mice were i.n. sensitized with HDM (*D. pteronyssinus*, 1 μg) on day 0 in 20 μL PBS. From days 7 to 11, these mice were i.n. challenged with 10 μg of HDM in 20 μL PBS without anesthesia. On day 11, the AR symptoms of mice were measured. In detail, after the last nasal drip, each mouse was placed alone in the cage

**Fig. 5 | Internal m⁷G-modified *Lrp1* mRNA is shuttled by QKI7 into SGs thus repressing its translation. a** BMDMs were administrated with HDM (200 μg/well), and relative *Lrp1* mRNA distribution in each ribosome fractions was analyzed by qRT-PCR ($n = 3$, biological replicates). (Tukey's HSD). **b** In BMDMs from *G3bp1^{f/f}* and *G3bp1^{mac-/-}* mice, relative *Lrp1* mRNA distribution in each ribosome fractions was analyzed by qRT-PCR ($n = 3$, biological replicates). (Tukey's HSD). **c** Sequencing read clusters from m⁷G MeRIP-seq analysis of *Lrp1* mRNA in PMs and top consensus motif identified by HOMER with MeRIP-seq peaks. **d** m⁷G modification of *Lrp1* mRNA in primary macrophages was examined by m⁷G-RIP-qRT-PCR ($n = 3$, biological replicates). (two-tailed unpaired Student's *t*-test with Welch's correction). **e** Immunofluorescence of G3BP1 and m⁷G-modified mRNA, and RNA-scope imaging of *Lrp1* mRNA in PMs. **f** Immunofluorescence of G3BP1 and QKI7, and RNA-scope imaging of *Lrp1* mRNA in PMs. **g** The association between QKI7 and *Lrp1* mRNA in primary macrophages was determined by RIP-qRT-PCR ($n = 3$, biological replicates). (two-tailed unpaired Student's *t*-test). **h** The association between the KH domain of QKI7 and *Lrp1* mRNA in RAW 264.7 cells was determined by RIP-qRT-PCR ($n = 3$, biological replicates). (Tukey's HSD). **i** The association between QKI7 and *Lrp1* mRNA in PMs was determined by RNA pull-down assay. **j** In QKI7-overexpressed RAW 264.7 cells stimulated by HDM (200 μg/well), relative *Lrp1* mRNA distribution in each ribosome fractions was analyzed by qRT-PCR ($n = 3$, biological replicates). (Tukey's HSD). Data are shown as mean ± s.d. or photographs from one representative of three independent experiments. Δ not significant.

position for 15 min, and the number of scratching and sneezing of each mouse within 15 min was counted. On day 12, these mice were anesthetized for further analysis.

Nasal lavage fluid (NLF) was collected. The nasal mucosa was harvested for hematoxylin-eosin (H&E) staining, periodic acid-Schiff (PAS) staining, flow cytometry assay, RNA isolation and immunofluorescence. Histological scores were calculated based on the status of epithelial cells: (0) normal, (1) loss of microvilli, (2) part shedding, (3) complete shedding; and cell infiltration: (0) normal, (1) mild, (2) severe, and (3) exudate to the lumen[45]. Lungs was harvested for H&E staining. For cell analysis, the nose was lavage three times with 1 mL cold PBS with 10 mM EDTA, and the lavage was centrifuged to collect the precipitates. For cytokine assessment, the nose was lavage three times with 0.5 mL cold PBS with 10 mM EDTA, and the supernatants were collected, then IL-4, IL-5, and IL-13 protein levels were measured by ELISA. Serum was collected and OVA-specific IgE levels was examined by ELISA.

### Cell culture and transfection
Mouse macrophage cell line RAW 264.7 was obtained from American Type Culture Collection (ATCC). Mouse macrophage iBMDM cell line was donated by Prof. Haipeng Liu from Shanghai Pulmonary Hospital[46]. RAW 264.7 and iBMDM cells were cultured in DMEM with 10% FBS. Jet-PRIME transfection reagents were used for the transfection of plasmids according to the manufacturer's instruction. Mouse peritoneal macrophages (PM) and bone marrow-derived macrophages (BMDM) were obtained as previously described[47,48]. PMs and BMDMs were transfected with siRNAs using Lipofectamine RNAiMAX according to the manufacturer's instruction. The mouse G3BP1 specific siRNA were 5′- GCA UCU GUG ACG AGU AAG AUU -3′ (sense) and 5′- UCU UAC UCG UCA CAG AUG CUU -3′ (antisense); the mouse QKI specific siRNA were 5′- GCA UCU AAA UGA AGA CUU AUU -3′ (sense) and 5′- UAA GUC UUC AUU UAG AUG CUU -3′ (antisense); the scrambled control RNA sequences were 5′-UUC UCC GAA CGU GUC ACG UTT-3′ (sense) and 5′-ACG UGA CAC GUU CGG AGA ATT-3′ (antisense). siRNA was synthesized from GenePharma (Shanghai, China). siRNA duplexes were transfected at a final concentration of 40 nM.

Mouse bone marrow neutrophils (BMN) were isolated from bone marrow, and then were enriched by EasySep™ Mouse Neutrophil Enrichment Kit according to the manufacturer's instruction. Mouse bone marrow-derived eosinophils (BMDE) were generated from bone marrow cells following the previously described protocol[30]. Briefly, bone marrow cells were cultured in RPMI 1640 supplemented with 20% FBS, 100 IU/mL penicillin/streptomycin, 2 mmol/L L-glutamine, 25 mmol/L HEPES, 1×nonessential amino acids, 1 mmol/L sodium pyruvate, and 50 mmol/L 2-mercaptoethanol. In the first 4 days, SCF (100 ng/mL) and FLT3 ligand (100 ng/mL) were added in the media, and in the remaining 10 days, the media was switched and IL-5 (10 ng/mL) was added. NMMs were isolated from nasal mucosa of OVA or HDM-induced AR mice, and then were enriched by EasySep Mouse F4/80 Enrichment Kit according to the manufacturer's instruction. Splenic T cells were isolated from spleens of *G3bp1^{f/f}* and *G3bp1^{mac-/-}* mice by

magnetic-activated cell sorting kit (EasySep Mouse T Cell Isolation Kit) according to the manufacturer's instruction. Naive CD4+ T cells were isolated from mouse spleens using a magnetic-activated cell sorting kit (CD4+ T Cell Isolation Kit, Miltenyi Biotec), and these T cells were activated with plate-coated anti-CD3 Ab (2 μg/mL) and anti-CD28 Ab (2 μg/mL). After 2 days, these cells were supplemented with mouse IL-4 (20 ng/mL) and anti-IFN-γ Ab (10 μg/mL) to induce Th2 activation following previously established protocols[49].

### In vitro efferocytosis assay
For the induction of apoptosis, human Jurkat cells were irradiated with 254-nm UV lamp for 15 min and then placed in the incubator for 2 h, BMDEs were cultured with 1 mmol/L budesonide in the absence of IL-5 overnight, and BMNs were cultured in HBSS containing 1% FBS overnight according to the previously reported protocols[30,50–52]. The apoptotic cells (AC) were resuspended and incubated for 5 min with 1 mL of Diluent C containing PKH67, PKH26, or CellVue Claret membrane-intercalating dyes. Cells were rinsed twice with PBS and resuspended. PMs or BMDMs were incubated with ACs at a 1:5 macrophage:AC ratio for 45 min. Phagocytosis was assessed by flow cytometry, Incucyte imaging, or immunofluorescent confocal microscopy. For analyzing the effect of stress on efferocytosis, PMs or BMDMs were pre-treated with 50 μM NaAsO₂ for 30 min, 100 μg HDM for 1 h, or 100 μM G3Ia for 20 min, co-cultured with ACs for 45 min, and then subjected to efferocytosis examination.

### In vivo efferocytosis assay
According to the reported protocol[50], mice were injected intraperitoneally with 2% w/v thioglycolate. After 72 h, animals received intraperitoneally 5 × 10⁶ ACs that were pre-stained with PKH26, together with 50 μM NaAsO₂ or 100 μg HDM. 3 h latter, mice were sacrificed and efferocytosis was analyzed in the peritoneal exudates with flow cytometry. Phagocytic macrophages were identified as PE⁺F4/80⁺ cells.

### Flow cytometry
Cells were isolated from the nasal mucosa of AR mice according to the protocol[53]. Briefly, the nasal mucosa of AR mice was separated and digested in the 2 mL buffer with collagenase I 5000 U/mL, hyaluronidase 11400 U/mL, and DNase I 20 μL for 1 h at 37 °C. The cells were filtered through a 40 μm cell strainer and then centrifuged to obtain the precipitates. Then cells were incubated with Fc Receptor Blocking Solution anti-mouse CD16/32 for 30 min. After centrifuge, cells were further incubated with different antibody panels. Panel 1: BV711-conjugated anti-CD45 antibody, PE-conjugated anti-CD11b antibody, FITC-conjugated anti-F4/80 antibody, APC-conjugated anti-SiglecF antibody, and Percp.cy5.5-conjugated anti-Ly6G antibody. Panel 2: BV711-conjugated anti-CD45 antibody, PE-conjugated anti-CD11b antibody, FITC-conjugated anti-CD3ε antibody, and APC-CY7-conjugated anti-CD19 antibody. Eosinophils are defined as CD11b⁺Ly6G⁻SiglecF⁺, neutrophils are defined as CD11b⁺Ly6G⁺, macrophages are defined as CD11b⁺Ly6G⁻F4/80⁺, T cells are defined as

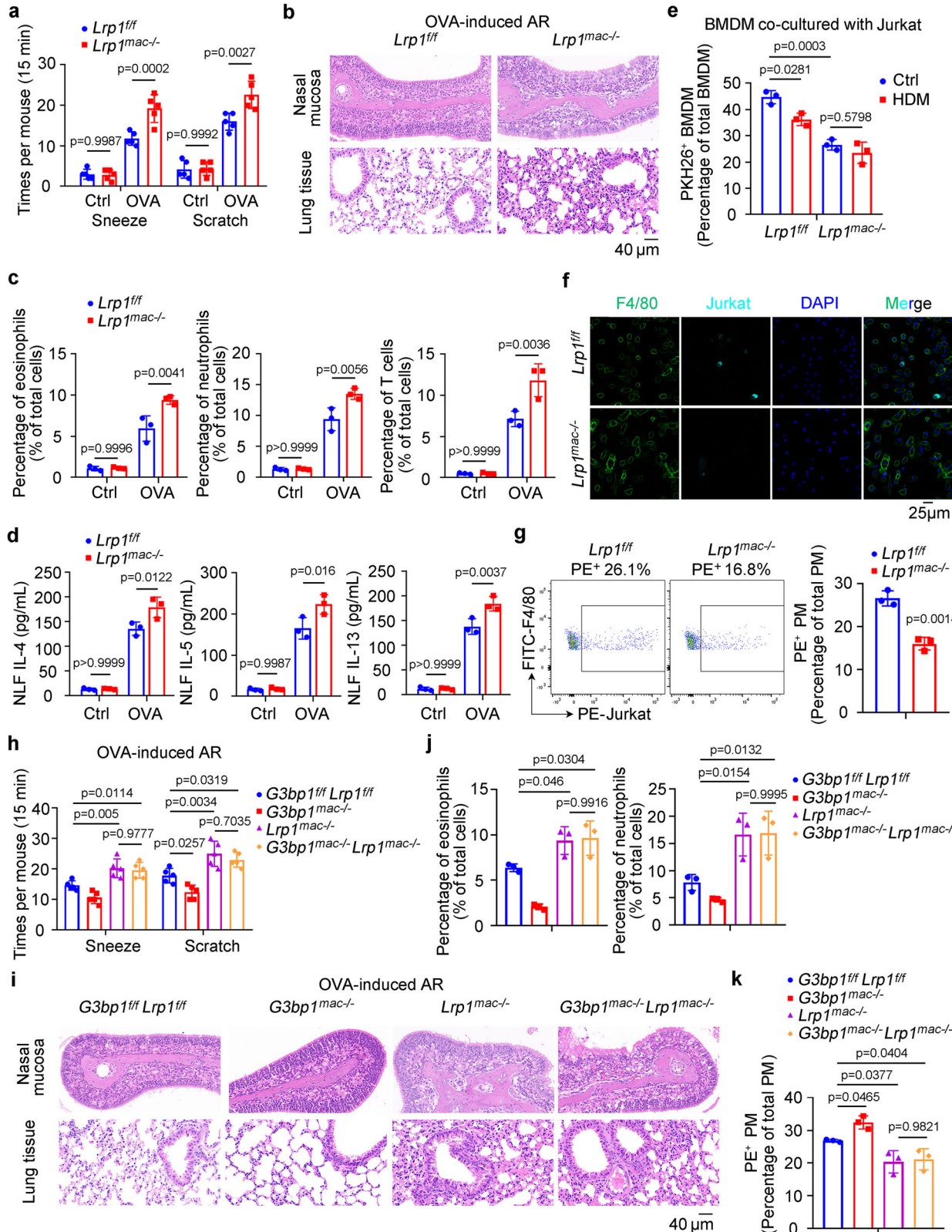

CD11b⁻CD3⁺, and B cells are defined as CD11b⁻CD3⁻CD19⁺. For flow cytometry analysis of efferocytosis, cells were incubated with Fc Receptor Blocking Solution anti-mouse CD16/32 for 30 min. After centrifuge, cells were further incubated with the fluorescent antibodies for 15 min to label cells respectively. The data were obtained from Fortessa flow cytometry and analyzed using FlowJo software (BD Bioscience, CA).

## H&E and immunofluorescence staining

Paraffin-embedded tissue sections were stained with H&E to visualize nasal mucosa pathology as previously described[54]. For immunofluorescence, sections were stained with primary antibodies against CD3, CD19, CD11b, G3BP1, or CD68. For cell immunofluorescence, macrophages were stained with primary antibodies against G3BP1, eIF4G, eIF3b, F4/80, QKI7, or m⁷G as previously described [53]. The slides

**Fig. 6 | The SG-promoted AR depends on the inhibition of efferocytosis receptor LRP1. a–d** *Lrp1*^*f/f*^ and *Lrp1*^*mac-/-*^ mice were administered with OVA to induce AR symptoms as depicted in Supplementary Fig. 2e. **a** Times of sneezes and scratches in each mouse was counted in 15 min after last i.n. challenge (*n* = 5, biological replicates). (Tukey's HSD). **b** H&E staining of the nasal mucosa and lung tissues from AR mice. **c** Flow cytometry of eosinophils, neutrophils and T cells in the nasal mucosa of control and AR mice (*n* = 3, biological replicates). (Tukey's HSD). **d** ELISA assay of IL-4, IL-5, and IL-13 in the NLF obtained from control and AR mice (*n* = 3, biological replicates). (Tukey's HSD). **e** In vitro efferocytosis assay of PKH26-labeled apoptotic Jurkat cells engulfed by *Lrp1*^*f/f*^ and *Lrp1*^*mac-/-*^ BMDMs was measured by flow cytometry (*n* = 3, biological replicates). (Tukey's HSD). **f** Representative immunofluorescence assay showing efferocytosis of apoptotic cells (labeled with Claret) by *Lrp1*^*f/f*^ and *Lrp1*^*mac-/-*^ macrophages (labeled with F4/80). **g** In vivo efferocytosis assay of PKH26-labeled apoptotic Jurkat cells engulfed by

*Lrp1*^*f/f*^ and *Lrp1*^*mac-/-*^ macrophages was measured by flow cytometry (*n* = 3, biological replicates). (two-tailed unpaired Student's test). **h–j** *G3bp1*^*f/f*^*Lrp1*^*f/f*^, *G3bp1*^*mac-/-*^, *Lrp1*^*mac-/-*^, and *G3bp1*^*mac-/-*^*Lrp1*^*mac-/-*^ mice were administered with OVA to induce AR symptoms as depicted in Supplementary Fig. 2e. **h** Times of sneezes and scratches in each mouse was counted in 15 min after last i.n. challenge (*n* = 5, biological replicates). (Dunnett's multiple comparisons test). **i,** H&E staining of the nasal mucosa and lung tissues from AR mice. **j** Flow cytometry of eosinophils and neutrophils in the nasal mucosa of control and AR mice (*n* = 3, biological replicates). (Dunnett's multiple comparisons test). **k** In vivo efferocytosis assay of PKH26-labeled apoptotic Jurkat cells engulfed by *G3bp1*^*f/f*^*Lrp1*^*f/f*^, *G3bp1*^*mac-/-*^, *Lrp1*^*mac-/-*^, and *G3bp1*^*mac-/-*^*Lrp1*^*mac-/-*^ macrophages was measured by flow cytometry (*n* = 3, biological replicates). (Dunnett's multiple comparisons test). Data are shown as mean ± s.d. or photographs from one representative of three independent experiments.

were mounted by Mounting Medium with DAPI. The pictures were captured using Leica MDI8 confocal microscope (North Deerfield, IL). The number of G3BP1 aggregates, cells with G3BP1 assembly in nasal mucosa, and TUNEL fluorescence ratio of nasal mucosa were quantified using Image J software.

### Fluorescence recovery after photobleaching

RAW 264.7 cells transfected with GFP-tagged G3BP1 were seeded on glass bottom dish (NEST, 801001) with or with stimulus, and examined with a confocal microscope (Leica DMI8). NMMs were also seeded on glass bottom dish, transfected with GFP-tagged G3BP1 using FuGENE HD, and examined with a confocal microscope with HDM stimulus. For FRAP experiment, a pulse bleached 70%-80% of the fluorescence in a rectangle region of interest (ROI) of G3BP1-GFP, and the recovery of fluorescence intensity was monitored.

### RNA extraction and real-time PCR

Total RNA was isolated from macrophages by TRIzol reagent according to the manufacturer's instruction. Real-time quantitative RT-PCR (qRT-PCR) was performed using SYBR RT-PCR kit (RR430B, Takara, Dalian, China) and LightCycler (Roche, Switzerland) as described previously [43]. The qPCR primers were mouse *G3bp1* (forward: 5′- TTG GAG GAG CAT TTA GAG GAG C -3′, reverse: 5′- TCT TGA ATG TCG GAC ACA GGT -3′); mouse *Il-4* (forward: 5′- GGT CTC AAC CCC CAG CTA GT -3′, reverse: 5′- GCC GAT GAT CTC TCT CAA GTG AT -3′); mouse *Il-5* (forward: 5′- GCA ATG AGA CGA TGA GGC TTC -3′, reverse: 5′- GCC CCT GAA AGA TTT CTC CAA TG -3′); mouse *Il-13* (forward: 5′- TGA GCA ACA TCA CAC AAG ACC -3′, reverse: 5′- GGC CTT GCG GTT ACA GAG G -3′); mouse *Il-6* (forward: 5′- TAG TCC TTC CTA CCC CAA TTT CC -3′, reverse: 5′- TTG GTC CTT AGC CAC TCC TTC -3′); mouse *Il-10* (forward: 5′- CTT ACT GAC TGG CAT GAG GAT CA -3′, reverse: 5′- GCA GCT CTA GGA GCA TGT GG -3′); mouse *Prg2* (forward: 5′- GTC TCA GGT CAG GAT GTG ACA -3′, reverse: 5′- GCG GAC TGG ATT CCG AAG TT-3′); mouse *Epx* (forward: 5′- TAG GGG CCT TAG CCA CAC TC −3′, reverse: 5′- CTG CTA TGC AGT CTC GAA GGA -3′); mouse *Mpo* (forward: 5′- AGG GCC GCT GAT TAT CTA CAT -3′, reverse: 5′- CTC ACG TCC TGA TAG GCA CA -3′); mouse *Lrp1* (forward: 5′- ACT ATG GAT GCC CCT AAA ACT TG -3′, reverse: 5′- GCA ATC TCT TTC ACC GTC ACA -3′); internal control mouse *β-actin* (forward: 5′- AGT GTG ACG TTG ACA TCC GT -3′, reverse: 5′- GCA GCT CAG TAA CAG TCC GC -3′); mouse *Gapdh* (forward: 5′- AGG TCG GTG TGA ACG GAT TTG-3′, reverse: 5′- GGG GTC GTT GAT GGC AAC A -3′). The relative expression of the individual genes was normalized to that of internal control using $2^{-\Delta\Delta Ct}$ cycle threshold method in each sample.

### Preparation of RNA samples from SGs

We prepared RNA samples from SGs as SG core-enriched fractions following a previous study [24]. After HDM stimulation for 1 h, macrophages were washed twice with PBS and harvested in 1.5 mL SG lysis buffer (50 mM Tris pH 7.6, 50 mM NaCl, 5 mM MgCl₂, 0.1% NP-40, 1 mM

β-mercaptoethanol, 1% EDTA-free protease inhibitor cocktail, and 0.4 U/mL RNase inhibitor) on ice for 20 min. Then, cell lysates were shaken and mixed. 1/10 of the cell lysates were taken as total cell fractions. The other cell lysates were centrifuged at 2000 g for 5 min at 4 °C. The cytosolic fraction was in the supernatant, and the nuclear fraction was in the pellet. After that, 1 mL supernatant was centrifuged at 18,000 g for 20 min at 4 °C to isolate the cytosolic solventless pellet fraction. The pellet was re-suspended in 1 mL SG lysis buffer and centrifuged at 18,000 g for 20 min at 4 °C again to purify the pellet components. Then, the supernatant was discarded, and the pellet was re-suspended in 300 mL SG lysis buffer and centrifuged at 850 g for 2 min at 4 °C. The supernatant represented SG core-enriched fractions. The RNAs from total cell fractions, and SG core-enriched fractions were extracted using TRIzol reagent.

### RNA-seq and data processing

Total RNA was extracted from macrophages by TRIzol reagent followed by library preparation according to Illumina standard instruction (VAHTS Universal V6 RNA-seq Library Prep Kit for Illumina). Agilent 4200 bioanalyzer was employed to evaluate the concentration and size distribution of cDNA library before sequencing with an Illumina novaseq6000. The protocol of high-throughput sequencing was fully according to the manufacturer's instructions (Illumina). The raw reads were filtered by Seqtk before mapping to genome using Hisat2 (version: 2.0.4). The fragments of genes were counted using stringtie (v1.3.3b) followed by TMM (trimmed mean of M values) normalization [55]. Significant differentially expressed genes (DEGs) were identified as those with a False Discovery Rate (FDR) value above the threshold ($q < 0.05$) and fold-change >2 using edgeR software [56].

### scRNA-seq and data processing

Mouse nasal mucosa was digested for single cell suspension as previously reported [53]. The dissociated single cells were stained with AO/PI staining for viability assessment using Countstar Fluorescence Cell Analyzer. The scRNA-seq libraries were generated using the 10X Genomics Chromium Controller Instrument and Chromium Single Cell 3′V3.1 Reagent Kits (10X Genomics, Pleasanton, CA) according to the manufacturer's recommendations. scRNA-seq data analysis was performed by NovelBio Bio-Pharm Technology Co.,Ltd, with NovelBrain Cloud Analysis Platform as we previously reported [17].

### RNA immunoprecipitation-seq and RIP-qPCR

Primary macrophages and RAW 264.7 cells seeded in a 15 cm dish at 85% confluence were cross-linked by ultraviolet light and harvested. RNA immunoprecipitation (RIP) was performed using the Magna RIP® RNA-Binding Protein Immunoprecipitation Kit (Millipore) according to the manufacturer's instruction. Cell lysates of macrophages were subjected to RIP by rabbit IgG that served as negative control or anti-G3BP1. Cell lysates of transfected-RAW 264.7 cells were subjected to RIP by mouse IgG that served as negative control or anti-Flag. Input

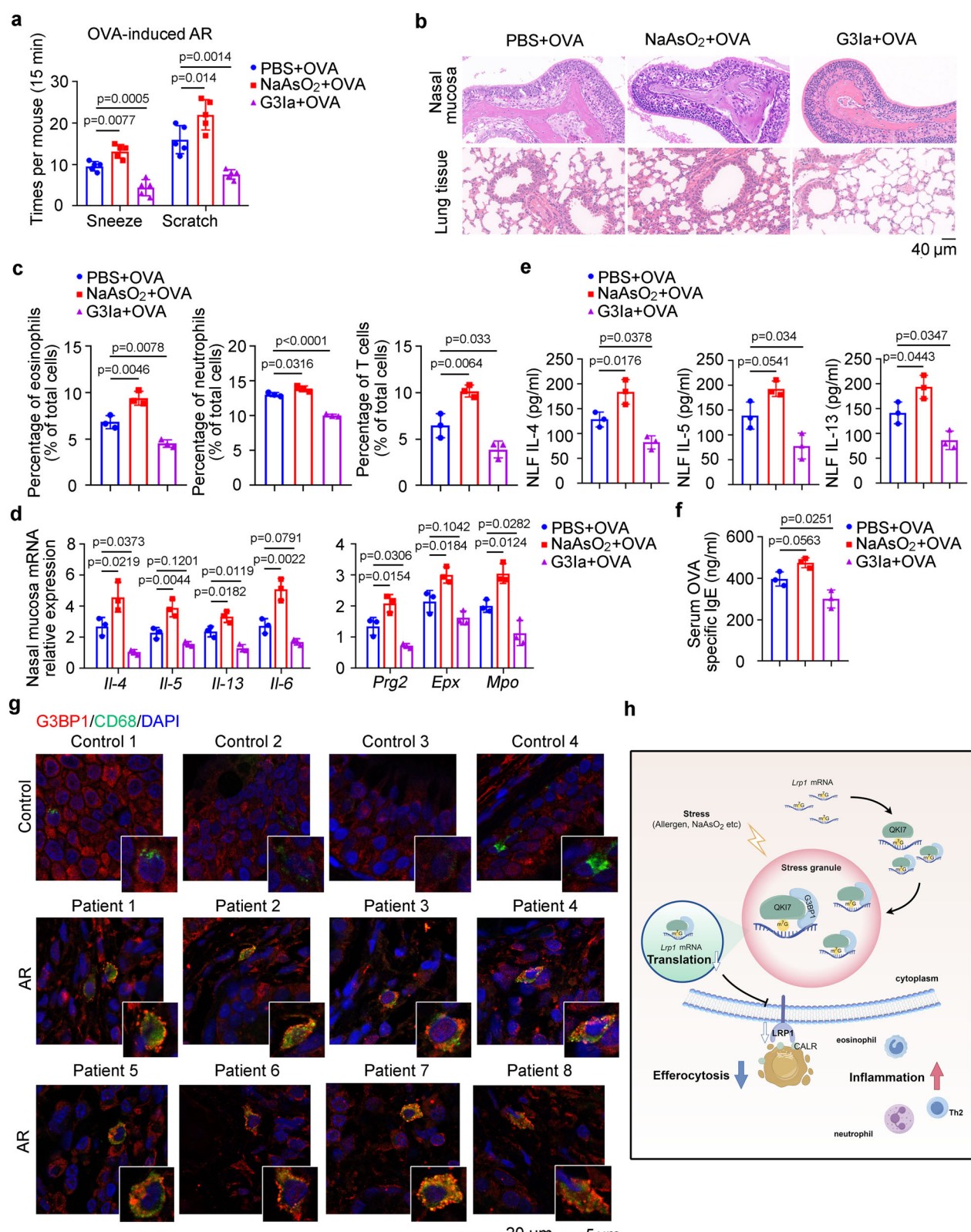

and co-immunoprecipitated RNAs were isolated by TRIzol for sequencing and qRT-PCR. RIP-seq was performed by Biotechnology Corporation (Shanghai, China).

## Decapping of mRNA

Dynabeads Oligo(dT)25 (61005, Thermo Fisher Scientific) was used to capture and purify mRNA from total RNA. The PolyA+ RNAs were treated with DNase at 37 °C for 30 min to remove DNA contamination. Decapping of mRNA was performed as a previously reported protocol using Tobacco Decapping Plus 2[24]. A maximum of 6 μg purified polyA+ RNAs were incubated with 5 μL 10× Decapping Reaction Buffer (100 mM Tris-HCl pH 7.5, 1.0 M NaCl, 20 mM MgCl₂, and 10 mM DTT), 1 μL 50 mM MnCl₂, 2 μL SUPERase-In RNase Inhibitor, and 8 μL Tobacco decapping enzyme in a final volume of 50 μL. The reaction

**Fig. 7 | SG assembly inhibitor alleviates AR symptoms, and SGs are assembled in nasal macrophages of AR patients. a–f** Mice were administrated by OVA and treated with PBS, NaAsO₂ (0.015 mg/kg), or G3Ia (0.05 mg/kg) as depicted in Supplementary Fig. 7b. **a** Times of sneezes and scratches in each mouse was counted in 15 min after last i.n. challenge ($n = 5$, biological replicates). (Dunnett's multiple comparisons test). **b** H&E staining of the nasal mucosa and lung tissues from AR mice. **c** Flow cytometry of eosinophils, neutrophils and T cells in the nasal mucosa of AR mice ($n = 3$, biological replicates). (Dunnett's multiple comparisons test). **d** ELISA assay of IL-4, IL-5, and IL-13 in the NLF obtained from AR mice ($n = 3$, biological replicates). (Dunnett's multiple comparisons test). **e** mRNA expressions of cytokines, *Prg2*, *Epx*, and *Mpo* were detected by qRT-PCR in the nasal mucosa of AR mice ($n = 3$, biological replicates). (Dunnett's multiple comparisons test). **f** Serum OVA-specific IgE level in the AR mice ($n = 3$, biological replicates). (Dunnett's multiple comparisons test). **g** Representative immunofluorescence imaging showing the SG assembly in macrophages (CD68) in the nasal mucosa from healthy controls and AR patients. **h** Working model for SG assembly-inhibited macrophage efferocytosis through *Lrp1* mRNA translational arrest to aggravate AR progression, created by Figdraw (export ID: IRIRWb0667). Data are shown as mean ± s.d. or photographs from one representative of three independent experiments.

was incubated at 37 °C for 2 h. Decapped mRNA was then purified from the solution using Oligo Clean & Concentrator.

## RNA-scope and m⁷G immunofluorescence staining

For the RNA in situ hybridization, we used the RNAscope Multiplex Fluorescent v2 Assay combined with Immunofluorescence (Advanced Cell Diagnostics, ACD, Newark, CA) according to the manufacturer's instructions. Specific probes against mouse *Cd45*, *Krt5* and *Lrp1* mRNA were designed by ACD. Briefly, paraffin-embedded nasal mucosa tissue sections were baked for 1 h, deparaffinized, and treated with hydrogen peroxide for 10 min. Then we performed target retrieval at 100 °C for 15 min and applied primary antibody diluted in Co-Detection Antibody Diluent on each section. Next day, sections were submerged in 10% Neutral Buffered Formalin (NBF) for 30 min at RT, and incubated with Protease Plus at 40 °C for 30 min into the HybEZ™ Oven. For cells, macrophages seeded on 8-well chamber slides were fixed by 10% NBF, pre-treated with Perm/Wash Buffer, and then applied primary antibody diluent overnight. After that, sections and cells were incubated with the appropriate probes for 2 h at 40 °C, and washed with washing buffer. Amplification was done by incubation at 40 °C with AMP (1-4) buffers for 30, 15, 30, and 15 min respectively, with two times 2 min wash between steps. Finally, sections were mounted by mounting medium with DAPI in the dark at 4 °C. Pictures were acquired with a Leica DMI8 confocal microscope. For m⁷G immunofluorescence staining, the in-situ mRNA decapping procedure was conducted using Tobacco Decapping Plus 2. Then, the m⁷G-immunostaining primarily detected mRNAs with internal m⁷G modification following the previously reported protocol[24].

## mRNA m⁷G MeRIP-seq and m⁷G MeRIP-qPCR

m⁷G-MeRIP-seq was conducted following the previously published protocol [24]. Total RNA was subjected to immunoprecipitation with the GenSeq m⁷G MeRIP Kit (GenSeq Inc.) following the manufacturer's instruction. Briefly, RNA was decapping using Tobacco Decapping Plus 2 enzyme, and then randomly fragmented to ~200 nt using NEBNext Magnesium RNA Fragmentation Module. Protein A/G beads were coupled to the m⁷G antibody by rotating at RT for 1 h. The RNA fragments were incubated with the bead-linked antibodies and rotated at 4 °C for 4 h. The RNA/antibody complexes were then digested with Proteinase K and the eluted RNA was purified by phenol:chloroform extraction. Both "input" group and "IP" group were prepared for next-generation sequencing or qRT-PCR. RNA libraries for IP and input samples were then constructed with GenSeq Low Input RNA Library Prep Kit (GenSeq, Inc.) following the manufacturer's instruction. Libraries were qualified using Agilent 2100 bioanalyzer (Agilent) and then sequenced in the sequencer. m⁷G MeRIP-seq service and the subsequent bioinformatics analysis were performed by CloudSeq Biotech Inc. (Shanghai, China).

## Western blotting

Cells were collected and lysed with cell lysis buffer (Cell Signaling Technology, 9803) supplemented with protease inhibitor cocktail. Protein concentrations of the lysates were measured using bicinchoninic acid (BCA) protein assay kit (Pierce, 23250) and equalized with cell lysis buffer. Equal amount of the extracts was loaded and subjected to SDS-PAGE, transferred onto nitrocellulose membranes, and then blotted as we described previously[43]. Protein levels were quantified using Image J software and β-actin was used as a loading control.

## RNA pull-down

The RNA oligos labeled with biotin targeting mouse *Lrp1* mRNA were constructed by Ribobio (Guangzhou, China). The probe sequence: probe_1 (5′- AAT CTC TTT CAC CGT CAC AC-/3bio/ -3′); probe_2 (5′- CCT CAT ATA CAA TCT TCC GG-/3bio/ -3′); probe_3 (5′- TCT TCT CTG ACA CCT GAT CT-/3bio/ -3′); probe_4 (5′- ACG TAG TCT TCA AAT AGG GT-/3bio/ -3′); probe_5 (5′- TCC TTC ATA CCT TTG AGC CA-/3bio/ -3′); Negative control probe (5′- TGG CTC AAA GGT ATG AAG GA-/3bio/ -3′). Then, RNA pull-down was conducted using Pierce™ Magnetic RNA Protein Pull Down Kit according to the manufacturer's instruction. Briefly, 50 pmol 3′ end biotin-labeled RNA oligos were bound with 50 μL streptavidin magnetic beads in RNA capture buffer for 1 h at RT with agitation. Streptavidin beads-conjugated RNA oligos were then incubated with 200 μg macrophages cell lysate in Protein-RNA binding buffer in 100 μL final volume overnight at 4 °C. Finally, the beads were washed with washing buffer for 4 times, and the RNA-protein complexes were eluted using Elution Buffer. Proteins were detected by Western blotting.

## Polysome profiling

The polysome profiling was conducted according to the reported protocol[17]. A 10 cm dish of BMDMs or transfected-RAW 264.7 cells at 80% confluence was used for the polysome profiling. Before collection, CHX was added to the medium at 100 mg/mL for 7 min to block active mRNA translation. Then the cells were washed with PBS containing 100 mg/mL CHX for three times. 500 mL lysis buffer (20 mM HEPES pH 7.6, 100 mM KCl, 5 mM MgCl₂, 100 mg/mL CHX, 1% Triton X-100, 1% protease inhibitor cocktail, and 40 U/mL RNase inhibitor) was added to lysate the cells on ice for 30 min, and the lysate was centrifuged at 15,000 g for 15 min at 4 °C. 400 mL supernatant was layered on top of a 10% to 50% w/v sucrose gradient and centrifuged at 4 °C for 90 min in a SW41Ti swinging bucket rotor at 250,000 g. After centrifuge, the sample was fractionated into 12 fractions by Gradient Station (Bio-Camp) equipped with an ECONO UV monitor (BioRad) and collected with a fraction collector (FC203B, Gilson). RNA was purified from fractions by TRIzol reagent for qRT-PCR analysis.

## Statistical analysis

Data are shown as mean ± s.d., and are tested for normal distribution using the Shapiro-Wilk test ($n < 50$) and for homogeneity of variance using F-test. For comparisons between two groups, an unpaired parametric Student's t-test is applied if the data are normally distributed and meet homogeneity of variance; otherwise, an unpaired nonparametric test, the Mann-Whitney U test, is utilized. For comparisons of more than two groups, one-way ANOVA (one-way ANOVA with Tukey's Honestly Significant Difference (HSD) test or one-way

ANOVA with Dunnett's multiple comparisons test) is applied in SPSS 17.0 (Chicago, IL). All presented *p*-values were two-sided, and $p < 0.05$ was regarded as statistically significant.

## Reporting summary

Further information on research design is available in the Nature Portfolio Reporting Summary linked to this article.

## Data availability

The scRNA-seq data have been deposited in the GEO database under accession code GSE266238. The RNA-seq data have been deposited in GEO database under accession codes GSE264134. The SG RNA-seq data have been deposited in GEO database under accession codes GSE263848. The G3BP1 RIP-seq data have been deposited in GEO database under accession codes GSE264137. The m7G MeRIP-seq data have been deposited in GEO database under accession codes GSE263851. Public available datasets accessed for use in this manuscript include GSE90869, GSE112276, GSE192847, GSE193036, GSE192844, GSE192845, and GSE192846. All the original unprocessed gels and images, the FACS data and all the original source data of figures have been deposited and available at the public research database Mendeley Data Reserved [https://data.mendeley.com/datasets/kxksmyc89v/5]. All data are included in the Supplementary Information or available from the authors, as are unique reagents used in this Article. The raw numbers for charts and graphs are available in the Source Data file (Figshare https://doi.org/10.6084/m9.figshare.29136137). Source data are provided with this paper.

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

## Acknowledgements

We thank Prof. Haipeng Liu from Shanghai Pulmonary Hospital for kindly providing iBMDM cell line, and Ms. Tingting Fang from our laboratory for technical assistance. This work was supported by grants from National Key Research and Development Program of China (2023YFC2505900 to J.H.), National Natural Science Foundation of China (92369106 to Y.Z., 92269204 to J.H., 82421005 to S.L., U23A20439 to H.L., 82171749 to Y.Z.), Military Outstanding Youth Program (2020QN06119 to J.H., 01-SWKJYCJJ07 to J.H., 23SWAQ53 to J.H.), Shanghai Rising-Star Program (24QA2711700 to Y.Z.), Youth Talent Project of Shanghai Municipal Health Commission (2022YQ072 to Y.Z.), and Fundamental Research Funds of the Central Universities (31920220107 to Z.Y.).

## Author contributions

Y.Z., Z.Y., Y.W., Y.D., and T.W. performed experiments and contributed equally for the whole study. Y.L., C.L., Y.L., Z.L., S.L., L.G., Y.F., T.L., K.J., L.Z., M.W., W.N., L.C., M.M., Y.W., and C.Z. provided reagents and performed experiments. H.L. provided human samples and analyzed the data. Y.Z. and J.H. analyzed the data and wrote the paper. J.H. designed and supervised the study. All authors read and approved the final manuscript.

## Competing interests

The authors declare no competing interests.
