## [Transparent Peer Review file · Nature Communications]

Stress granule assembly impairs macrophage efferocytosis to aggravate allergic rhinitis in mice

Corresponding Author: Professor Jin Hou

Version 0:

Reviewer comments:

Reviewer #1

(Remarks to the Author)

This is an interesting work reporting formation of Stress Granules (SGs) in the nasal tissues from both allergic rhinitis (AR) mouse models and patients. Authors found that SGs were assembled specifically in the macrophages within nasal mucosa, which restrained the efferocytotic ability of macrophages and suppressed the clearance of apoptotic immune cells, thus promoting AR progression.

The study is well designed and conducted. I mostly agree with authors conclusions except few minor issues:

1. The FRAP analysis is not sufficient to conclude on the nature of LLPS. I would tone down this part especially because it is not the matter of the work
2. I would like to see more SG markers to be analyzed, especially eIF3b to conclude whether these SG are canonical or not
3. Please also use FISH to show accumulation of mRNAs into SGs of macrophages
4. Rescue experiments: compare wt and delta-NTF2 domain G3BP1 variants in G3BP1 KO cells.
5. What happen with G3BP2 when G3BP1 is depleted? Is there any compensation? What is G3BP2 expression profile in macrophages in general?

Reviewer #2

(Remarks to the Author)

The manuscript by Zhou et al. describes a novel role for stress granules and their major protein constituent G3BP1 in regulating LRP1-mediated macrophage efferocytosis and consequently Type 2 cytokine production and allergic rhinitis. The authors suggest an interesting paradigm with a detailed molecular dissection. However, some overreaching conclusions and falls in methodology reduce the enthusiasm toward this study and prevent publication at this stage. This can be changed if the following issues are rectified.

Major comments:

1. One of the major issues in this manuscript is the analysis of the molecular pathways in BMDM or peritoneal macrophages, or RAW cells and the projection to function of nasal mucosal cells. All the properties attributed to macrophages should be examined in NMM.
 2. There is also lack in forming a connection between efferocytosis and cytokine production by macrophages. In Suppl fig 8, the macrophages depicted for analysis are only F4/80hi. The monocytes (CD11bhi/F4/80lo) and Mres (CD11blow/F4/80low) should be analyzed as well. Also, Mres are generated following efferocytosis and their levels could be reduced following LRP1 reduction (PMID: 31375662; PMID: 31375662). Moreover, these macrophages produce IFNbeta that is critical for efferocytosis and the production of IL-10 during the resolution of inflammation. IL-10 is well-established anti- Th2 cytokine, and this pathway may explain the effects found in mice with reduced stress granules and reduced AR.
 3. In Figure 1a-c, the results of G3BP1 expression in nasal macrophages are not very convincing. Quantification of G3BP1 colocalization with CD11b should be done and compared between control and AR. Also, the WB results should show the cell populations presented by scRNA sequencing rather than macrophage cell lines.
- In addition, it is not clear what is the threshold for G3BP1+ macrophages in panel F since panel E of Suppl Fig. 1 shows very little G3BP1 expression in all cells. The photo bleaching and aggregations assays in response to HDM should be done in nasal macrophages and T cells (as controls) and not in macrophages of non-relevant location. Proper controls without

treatment should be shown with quantification (in Suppl Fig.1 as well).

4. In Figure 2 it should be stated which information is analyzed in which panel. Also, how panels b and c show better mucosa integrity and reduced mucus production should be detailed and quantified (also in figure 6). In addition, how does a 3-7% reduction in eosinophil frequency (panel D) results in a 2-3-fold reduction in total numbers is perplexing. Supp fig 2K should be quantified as well.
5. The results regarding the effect of stress granules on efferocytosis are clear, however, are they specific for apoptotic cells? What happens if you use latex beads of opsonized latex beads. Also, is it true for non-resident macrophages or AR-derived macrophages? Also, does the improved efferocytosis in mac G3BP1 KO results in better production of IL-4/IL-5, as previously reported for efferocytic macrophages (PMID: 23426944). That would counter the main paradigm.
6. The reduction in LRP1 expression following SG induction and the increase in G3BP1 Mac-KO seems marginal and likely could not explain the difference in efferocytosis. Please use LRP1 blockers to show a specific inhibition of efferocytosis and its impact on cytokine production.
7. In suppl Fig. 6 the expression of G3BP1 and LRP1 should be examined in nasal macrophages.
8. In figure 7, mice were not treated with G3la alone. This should be done to show the effect of G3BP1 inhibitor on "standard" AR. Also, the effect of HDM on OVA-induced AR and the impact of G3BP1 inhibitor was not examined. This should be done to strengthen the indicate involvement of stress granules and G3BP1 in AR.

Minor comments:

1. Please correct PHK26 to PKH26 throughout the manuscript.
2. In all WBs quantification of relevant bands should be added.
3. In the headline of Supp Fig. 4 internalizes should be replaced with sequesters.
4. In fig 7g, quantification of the number of nasal mucosa macrophages, and the number of macrophages with G3BP1 aggregation should be provided. In addition, it is not clear what do the different frames represent. The control on the left has much more staining of G3BP1 than the three frames to its right and no separated staining of G3BP1 and CD68 is shown. This is very confusing.
5. The scheme in Fig 7h shows NETosing eosinophils and neutrophils. The data in the manuscript suggests increased infiltration of live eosinophils and neutrophils and not NETosis. Please correct the figure. Also, the difference of this panel from Suppl fig 7f is not clear.
6. On Line 910, the protocol number and year of approval should be indicted.
7. On line 1033 the units of NAA are missing. On line 1036, the units of IL-5 seem incorrect..

Reviewer #3

(Remarks to the Author)

In this study, the authors identified a role of the assembly of cytoplasmatic stress granule (SG) in macrophages for aggravating allergic rhinitis (AR). They attributed this role to a decrease in efferocytosis by the macrophages. In addition, they found that SG assembly in macrophages led to a decrease in the amounts of Lrp1 protein in macrophages, due to repressed translation. Inhibiting SG assembly by genetic knockout or a G3BP1-specific inhibitor alleviated AR symptoms with increased LRP1 protein levels and rescued efferocytosis in macrophages. This works provides evidence for a possible mechanism of the role for macrophages in AR and a potential treatment target.

Major points

- 1) From 2h, it seems the SG formation is indicated in Th2 priming as well as AR responses at the local tissue. Thus, whether the former is dampened, precluding subsequent changes in the local tissue becomes a key question. The authors should show that 1) after i.p., but prior to i.n OVA-IgE titers are the same in ctrl and OVA-treated animals (same for S6G), and that 2) Th2 biasing with IL-4 + anti-CD3 or similar (in vitro) is unperturbed in the G3BP1^{mac-/-} line and LRP1^{mac-/-} line.
- 2) Given that two MO targeting cre lines have been used to show a positive and negative impact on AR, the impact of the MO cre's on disease needs to be shown as null, or if the cre targeting alone modifies disease outcomes. That is, c.f. Lyz2-cre, CSFR1cre to wild type in at least one model.
- 3) 3f. Assay is crude and susceptible to the spurious population coming through at the lower end of the F4/80 gate. Please also report PE^{hi} frequency by adjusting the PE⁺ gate to 5k on the PE-Jurkat axis. Similar for J, where part of the negative population is included in the gate. A no-jurkat control should be included in the assay, and ideally, unlabeled Jurkat cells or BMDE used. MFI of +ve fraction may further be a helpful metric in establishing veracity of outcome and sensitivity of assay to reveal differences.
- 4) Legends throughout should state statistical tests used in panels. E.g. S3F the test type is impt for small differences and needs to be stated. It should not be a t-test (as used throughout), approach should be revised after consulting a statistician. This may mean the data need normalization and then combining across the multiple experiments for robust interpretation, or the conclusions reframed. Also need to state biological replicate # vs technical replicates as not clear currently throughout.
- 5) Fig S2b,c. The specificity to myeloid lineage should be demonstrated. Show a non-macrophage cell type (e.g. T cells) for G3bp1 expression.
- 6) Fig S2F. The vehicle control is missing. Should have the vehicle treated control to show that there is increase AR symptoms.
- 7) Figure 4 title states that SG assembly internalizes Lrp1 mRNA to suppress its translation. Stronger evidence for this is needed to make this claim as it could equally be enhanced protein degradation. While Fig S5c/d speak to no change in degradation, it should be verified by FACS as in 4g,h and presented in the supplement alongside the Western blots. A confirmation that translation was inhibited with MG132 and CQ would also strengthen the claim.
- 8) Fig6h, the baseline sneeze and scratch responses in vehicle treated mice across the strains should be presented. If zero, this should be stated.
- 9) Fig 7G. The control should include CD68⁺ cells; at the moment it is simply showing a lack of MO in the control, which is different from SG formation in the disease state specifically. Further, given this is the human link, 7G should be quantified i.e.

SG/mm2 or similar in the two contexts.

Minor

- 1) The introduction should be simplified. It should state the rationale for the study, but at the moment is quite leading towards the outcome. Paragraph 1 and 2 could be integrated, and the leading sentence L105-106 should remove 'we presume' and be rewritten more agnostically. The manuscript would benefit from parsing by an English editing service.
- 2) Fig 1a The G3BP1 imaging was with poor quality for the identification of SG assembly. Some means of quantifying granules in a field may make this more robust (SG/mm2 or similar).
- 3) For the mouse models and measurements of sneezing/scratching, more detail is required. That is, clarify whether or not the mice were anaesthetized for i.n.s.
- 4) I couldn't find 'extended Fig 4A-c' should this be 'supplementary Fig 4A-C'?
- 5) L305: For the DKO, what is the cre driver? Both or just one. Not clear from text or methods.
- 6) What are the invading cells evidenced in 2B in the G3PB1Mac^{-/-} line? Everything Th2 associated in the mice is down in the rest of the figure, so what goes up to explain the increase in nuclei needs to be demonstrated. For e.g. are they CD45+ cells?

Reviewer #4

(Remarks to the Author)

Version 1:

Reviewer comments:

Reviewer #1

(Remarks to the Author)

All my issues were addressed adequately. I recommend this work for publication.

Reviewer #2

(Remarks to the Author)

Most of my concerns were addressed. However, the results in supplemental figure 2l are very important and should be moved to figure 2. Also, the increased expression of IL-10 in g3bpMac^{-/-} macrophages is important and should be examined in isolated nasal/peritoneal macrophages and tissue samples from nasal mucosa from g3bpMac^{-/-} mice during disease. The text should also indicate these findings. The abstract should be amended to "suppressed the clearance of apoptotic immune cells resulting in reduced Mres generation and IL-10 production...".

Moreover, the sentence "Next, less cytokines expression in the nasal mucosa (Fig. 2f and Supplementary Fig. 2m), and lower cytokines production in NLF of G3bp1 mac^{-/-} mice than those of controls were determined (Fig. 2g)" should be clarified and clearly indicate that the differences in IL-4, IL-5, IL-13, IL-6, Prg2, Epx, and Mpo (mRNA or protein) were found only in challenged mice and not their unchallenged controls.

Reviewer #3

(Remarks to the Author)

The response letter addresses concerns raised. To improve the manuscript quality in a further revision round, i) all data supplements should be referenced in the body text, and ii) incorporating the Th2 differentiation data presented in the response letter but not currently in the main document into a supplementary would improve veracity.

Reviewer #4

(Remarks to the Author)

Version 2:

Reviewer comments:

Reviewer #2

(Remarks to the Author)

No further comments. The manuscript can be accepted.

Reviewer #3

(Remarks to the Author)

Thank you for incorporating the references to the supplementary figures.

Reviewer #4

(Remarks to the Author)

To Reviewer #1:

Remarks to the Author:

This is an interesting work reporting formation of Stress Granules (SGs) in the nasal tissues from both allergic rhinitis (AR) mouse models and patients. Authors found that SGs were assembled specifically in the macrophages within nasal mucosa, which restrained the efferocytotic ability of macrophages and suppressed the clearance of apoptotic immune cells, thus promoting AR progression. The study is well designed and conducted. I mostly agree with authors conclusions except few minor issues.

Response: We sincerely appreciate your positive evaluation of our work as “well designed and conducted”. Regarding the “minor issues” you raised, we have now provided detailed point-by-point responses supported by new data. We believe that these revisions adequately address your concerns in the updated manuscript. Thank you very much for your valuable feedback and consideration.

Question 1: *The FRAP analysis is not sufficient to conclude on the nature of LLPS. I would tone down this part especially because it is not the matter of the work.*

Response: Thank you for your valuable comments. In our previous submission, we detected phase separation because it constitutes the essence of stress granules (SGs) assembly. Following your comments, we have now tone down this part and moved the relevant data to Supplementary Figure 1j and **new Supplementary Figure 1k**.

Question 2: *I would like to see more SG markers to be analyzed, especially eIF3b to conclude whether these SG are canonical or not.*

Response: We have now employed immunofluorescence to detect eIF3b, another SG marker, in HDM-stimulated macrophages, and confirmed that the assembled SGs are canonical (**new Supplementary Figure 1i**).

Question 3: *Please also use FISH to show accumulation of mRNAs into SGs of macrophages.*

Response: Following your comments, we have now used FISH to demonstrate the

accumulation of *Lrp1* mRNA in SGs of macrophages (**new Supplementary Figure 4k**). Compared with FISH, RNAscope, a newer RNA ISH technology, exhibited superior signal amplification and specificity (Figure 4d).

Question 4: Rescue experiments: compare wt and delta-NTF2 domain G3BP1 variants in G3BP1 KO cells.

Response: In response to your comments, we now examined *Lrp1* mRNA by RIP-qRT-PCR in *G3bp1^{mac-/-}* macrophages transfected with G3BP1 variants (**new Supplementary Figure 4i**). The efferocytotic ability of these transfected *G3bp1^{mac-/-}* macrophages was also analyzed (**new Supplementary Figure 4j**). These rescue experiments demonstrated that the NTF2 domain of G3BP1 is essential for its interaction with *Lrp1* mRNA and for inhibiting macrophage efferocytosis.

Question 5: What happen with G3BP2 when G3BP1 is depleted? Is there any compensation? What is G3BP2 expression profile in macrophages in general?

Response: We analyzed the *G3bp2* expression in immune cells using our scRNA-seq data (**Figure 1a-c for Reviewer**). We also detected G3BP2 protein level in *G3bp1^{mac-/-}* macrophages and observed no compensatory upregulation (**Figure 1d for Reviewer**), indicating that G3BP2 expression remains unchanged upon G3BP1 depletion.

Figure 1 for Reviewer. a. UMAP visualization of CD45⁺ cells in the nasal mucosa of

control and OVA-induced AR mice, with each dot representing a single cell and colored according to its cluster identity. **b.** UMAP visualization highlighting *G3bp2* expression within the CD45⁺ cells from nasal mucosa. **c.** Violin plot depicting *G3bp2* expression level in neutrophils, T cells, NK cells, macrophages, DCs, plasma cells, B cells and epithelial cells. **d.** G3BP2 protein level in BMDMs from *G3bp1^{ff}* and *G3bp1^{mac-/-}* mice.

To Reviewer #2:

Remarks to the Author:

The manuscript by Zhou et al. describes a novel role for stress granules and their major protein constituent G3BP1 in regulating LRP1-mediated macrophage efferocytosis and consequently Type 2 cytokine production and allergic rhinitis. The authors suggest an interesting paradigm with a detailed molecular dissection. However, some overreaching conclusions and falls in methodology reduce the enthusiasm toward this study and prevent publication at this stage. This can be changed if the following issues are rectified.

Response: We sincerely appreciate your positive evaluation of our work as “*novel*” and “*interesting*”. Regarding the concerns you raised, we have now provided detailed point-by-point responses supported by new data. We believe that these revisions adequately address your concerns in the updated manuscript. Thank you very much for your valuable feedback and consideration.

Major comments:

Question 1: *One of the major issues in this manuscript is the analysis of the molecular pathways in BMDM or peritoneal macrophages, or RAW cells and the projection to function of nasal mucosal cells. All the properties attributed to macrophages should be examined in NMM.*

Response: Following your comments, we have now isolated nasal mucosa macrophages (NMMs) from HDM-induced AR mice, and examined their efferocytosis. Our results demonstrate that G3BP1 deficiency in NMMs leads to increased efferocytotic ability and LRP1 protein expression (**new Supplementary Figure 3i and 4o**), confirming the suppressed macrophage efferocytosis by stress granule (SG) assembly in the nasal mucosa.

Question 2: There is also lack in forming a connection between efferocytosis and cytokine production by macrophages. In Suppl Fig 8, the macrophages depicted for analysis are only F4/80^{hi}. The monocytes (CD11b^{hi}/F4/80^{lo}) and Mres (CD11b^{low}/F4/80^{low}) should be analyzed as well. Also, Mres are generated following efferocytosis and their levels could be reduced following LRP1 reduction (PMID: 31375662; PMID: 31375662). Moreover, these macrophages produce IFN β that is critical for efferocytosis and the production of IL-10 during the resolution of inflammation. IL-10 is well-established anti-Th2 cytokine, and this pathway may explain the effects found in mice with reduced stress granules and reduced AR.

Response: Regarding to your comments, we have now quantified the percentages of resolution macrophages (Mres-F4/80^{med}CD11b^{low}) and monocytes (Mo-F4/80^{low}CD11b^{high}) following in vitro efferocytosis using flow cytometry. The results indicate that G3BP1 deficiency enhances the generation of Mres (**Figure 2a and b for Reviewer, new Supplementary Figure 8c**), while having no significant effect on monocytes. We also examined the *Il-10* mRNA levels in peritoneal macrophages co-cultured with or without apoptotic Jurkat cells, and found that macrophages from *G3bp1^{mac-/-}* mice exhibited increased expression of IL-10 compared to those from *G3bp1^{fl/fl}* mice (**Figure 2c for Reviewer**). Moreover, flow cytometry analysis revealed an increased generation of Mres in the nasal mucosa of *G3bp1^{mac-/-}* mice during AR (new Supplementary Figure 2I).

Figure 2. a. Representative flow cytometry plots illustrating the gating strategy for Mres and Mo following in vitro efferocytosis. **b.** Flow cytometry analysis of Mres and Mo post in vitro efferocytosis (n=3). **c.** The expression levels of *Il-10* mRNA in peritoneal macrophages following in vitro efferocytosis were examined by qRT-PCR (n=3). *, p<0.05; **, p<0.01, Δ, not significant.

Question 3: *In Figure 1a-c, the results of G3BP1 expression in nasal macrophages are not very convincing. Quantification of G3BP1 colocalization with CD11b should be done and compared between control and AR. Also, the WB results should show the cell populations presented by scRNA sequencing rather than macrophage cell lines. In addition, it is not clear what is the threshold for G3BP1+ macrophages in panel F since panel E of Suppl Fig. 1 shows very little G3BP1 expression in all cells. The photo bleaching and aggregations assays in response to HDM should be done in nasal macrophages and T cells (as controls) and not in macrophages of non-relevant location. Proper controls without treatment should be shown with quantification (in Suppl Fig.1 as well).*

Response: In response to your comments, we have now incorporated the corresponding controls and quantified the number of CD11b⁺ cells with SG assembly per 10 random fields (**new Supplementary Figure 1a and b**).

In Figure 1i, we previously examined G3BP1 protein expression in primary macrophages, neutrophils and eosinophils. Here, we have now isolated nasal mucosa macrophages (NMMs), nasal mucosa neutrophils (NMNs), and nasal mucosa CD3⁺ T cells (NMTs) from HDM-induced AR mice. G3BP1 protein levels were then examined, and it was highly expressed in NMMs (**new Figure 1j**).

Based on the standardized and normalized scRNA-seq data, cells with a *G3bp1* expression value greater than 0 were previously defined as *G3bp1*⁺ macrophages (Figure 1f). Here, we have updated the expression profile of *G3bp1* to clearly demonstrate its presence in nasal mucosa macrophages (**new Supplementary Figure 1g**).

We also stimulated primary T cells (NMTs) with HDM and examined SG formation, observing significantly less pronounced G3BP1 aggregation compared to that in macrophages (**new Supplementary Figure 1h**). In NMMs, we transfected GFP-tagged G3BP1 and observed SG assembly using FRAP assay (**new Supplementary**

Figure 1k and new Supplementary Video 3). Together, G3BP1 is expressed in NMMs, and SGs are assembled in these cells during AR.

Question 4: *In Figure 2 it should be stated which information is analyzed in which panel. Also, how panels b and c show better mucosa integrity and reduced mucus production should be detailed and quantified (also in figure 6). In addition, how does a 3-7% reduction in eosinophil frequency (panel D) results in a 2-3-fold reduction in total numbers is perplexing. Supp fig 2K should be quantified as well.*

Response: Following your comments, we have now detailed the analysis information for each panel in Figure 2 within the corresponding figure legends. We quantified the histological damage using histological damage scoring (**new Supplementary Figure 2i, 6d, and 6j**). Additionally, we quantified PAS⁺ goblet cells in the nasal mucosa using ImageJ software across 10 randomly selected fields (**new Supplementary Figure 2j**). We also quantified the TUNEL fluorescence ratio using ImageJ software across 10 random fields (**new Supplementary Figure 2o**).

In Figure 2d, the SiglecF⁺ ratio in the flow cytometry plot (number of SiglecF⁺CD45⁺CD11b⁺Ly6G⁻ cells relative to the number of CD45⁺CD11b⁺Ly6G⁻ cells) represents the percentage of eosinophils within the gated population, according to the traditional flow cytometry labeling techniques, which seems a 3-7% reduction in eosinophils frequency. The percentage of eosinophils relative to total cells (number of SiglecF⁺CD45⁺CD11b⁺Ly6G⁻ cells relative to the number of single live cells) more accurately reflects the extent of eosinophil infiltration, as statistically analyzed in the histogram on the left.

Question 5: *The results regarding the effect of stress granules on efferocytosis are clear, however, are they specific for apoptotic cells? What happens if you use latex beads of opsonized latex beads. Also, is it true for non-resident macrophages or AR-derived macrophages? Also, does the improved efferocytosis in mac G3BP1 KO results in better production of IL-4/IL-5, as previously reported for efferocytotic macrophages (PMID: 23426944). That would counter the main paradigm.*

Response: In response to your comments, we investigated the efferocytotic ability using latex beads (L3030, Sigma-Aldrich) in PMs and NMMs, representing non-resident macrophages and AR mucosa macrophages, and found that HDM-induced SG

assembly had little impact on the efferocytosis of latex beads (**Figure 3a for Reviewer**). This suggests that the effect of SG formation on efferocytosis may be specific to apoptotic cells, which express calreticulin (CALR) on their surface as a ligand for LRP1 recognition, and LRP1 is determined to be suppressed by SG assembly in this study.

We examined the mRNA expression of *Il-4* and *Il-5* in PMs co-cultured with or without apoptotic Jurkat cells, and found that their levels were similar between *G3bp1^{ff}* and *G3bp1^{mac-/-}* macrophages (**Figure 3b for Reviewer**). Furthermore, we measured the protein level of IL-4 in the supernatant, and found no significant difference (**Figure 3c for Reviewer**). Therefore, the enhanced efferocytosis observed in G3BP1-deficient macrophages is not attributable to the altered production of IL-4 and IL-5.

Figure 3 for Reviewer. **a.** In vitro efferocytosis of PMs and NMMs using latex beads was measured by flow cytometry (n=3). **b.** mRNA expression levels of *Il-4* and *Il-5* were measured by qRT-PCR in the *G3bp1^{ff}* and *G3bp1^{mac-/-}* PMs co-cultured with or without apoptotic Jurkat cells (n=3). **c.** ELISA assay of IL-4 protein level in the supernatant of *G3bp1^{ff}* and *G3bp1^{mac-/-}* PMs co-cultured with or without apoptotic Jurkat cells (n=3). Δ , not significant.

Question 6: *The reduction in LRP1 expression following SG induction and the increase in G3BP1 Mac-KO seems marginal and likely could not explain the difference in efferocytosis. Please use LRP1 blockers to show a specific inhibition of efferocytosis and its impact on cytokine production.*

Response: Regarding to your comments, we have now used recombinant mouse LRPAP protein, a LRP1 blocker, to treat PMs and evaluate efferocytosis, and confirmed the inhibition of macrophage efferocytosis by the suppression of LRP1 (**new Supplementary Figure 6g**). The inhibited macrophage efferocytosis we also determined in *Lrp1^{mac-/-}* mice in vivo (Figure 6g). Additionally, *Il-10* mRNA expression in PMs post efferocytosis was also inhibited by treatment with the LRP1 blocker LRPAP (**new Supplementary Figure 6h**).

Question 7: *In suppl Fig. 6 the expression of G3BP1 and LRP1 should be examined in nasal macrophages.*

Response: Following your comments, the protein levels of G3BP1 and LRP1 in NMMs were measured using Western Blot, thereby confirming the knockout efficiency (**new Supplementary Figure 6b**).

Question 8: *In figure 7, mice were not treated with G3Ia alone. This should be done to show the effect of G3BP1 inhibitor on “standard” AR. Also, the effect of HDM on OVA-induced AR and the impact of G3BP1 inhibitor was not examined. This should be done to strengthen the indicate involvement of stress granules and G3BP1 in AR.*

Response: In response to your comments, we have now evaluated the effects of G3Ia treatment alone on mice, and found that G3Ia did not alter the basal phenotypes, including symptoms and histological features (**new Supplementary Figure 7c and d**). Additionally, in HDM-induced AR mice, G3Ia treatment alleviated AR symptoms and reduced nasal mucosa damage (**new Supplementary Figure 7e and f**), confirming that G3Ia inhibits SG assembly and thereby suppresses AR.

Minor comments:

Question 1: *Please correct PHK26 to PKH26 throughout the manuscript.*

Response: We apologize for this mistake, and have now corrected PHK26 to PKH26 throughout the manuscript.

Question 2: *In all WBs quantification of relevant bands should be added.*

Response: Following your requirements, we have now quantified all the Western blot bands using ImageJ software, and have added them in the corresponding figures.

Question 3: In the headline of Supp Fig. 4 internalizes should be replaced with sequesters.

Response: Thank you for your comments, and we have now made this revision.

Question 4: In fig 7g, quantification of the number of nasal mucosa macrophages, and the number of macrophages with G3BP1 aggregation should be provided. In addition, it is not clear what do the different frames represent. The control on the left has much more staining of G3BP1 than the three frames to its right and no separated staining of G3BP1 and CD68 is shown. This is very confusing.

Response: The numbers of CD68⁺ cells and CD68⁺ cells with SG assembly per 10 random fields are now quantified (**new Supplementary Figure 7j and k**).

Each frame stands for a clinical sample and is now labeled accordingly (**new Figure 7g**). The increased staining of G3BP1 in Control 1 on the left may be attributed to inter-individual variability. Due to space constraints, we did not include the separate staining of G3BP1 and CD68 in these figures; however, they are provided below for your reference (**Figure 4 for Reviewer**).

Figure 4 for Reviewer. Representative IF images showing the SG assembly in macrophages (CD68) in the nasal mucosa from healthy controls and AR patients.

Question 5: *The scheme in Fig 7h shows NETosing eosinophils and neutrophils. The data in the manuscript suggests increased infiltration of live eosinophils and neutrophils and not NETosis. Please correct the figure. Also, the difference of this panel from Suppl fig 7f is not clear.*

Response: We sincerely appreciate your insightful comments, and have now revised the working model accordingly (**new Figure 7h**). The previous Supplementary Figure 7f is now integrated into the **new Figure 7g** to avoid potential misunderstanding.

Question 6: *On Line 910, the protocol number and year of approval should be indicated.*

Response: The ethics approval number for collecting human samples (2024SL128) is now added.

Question 7: *On line 1033 the units of NAA are missing. On line 1036, the units of IL-5 seem incorrect.*

Response: The Non-essential Amino Acids (NAA) solution (100×, Gibco, 11140076) was added to the culture medium at 1:100 for use, and this information is now added. We apologize for the mistake of IL-5 units, and the correct concentration is 10 ng/mL.

To Reviewer #3:

Remarks to the Author:

In this study, the authors identified a role of the assembly of cytoplasmic stress granule (SG) in macrophages for aggravating allergic rhinitis (AR). They attributed this role to a decrease in efferocytosis by the macrophages. In addition, they found that SG assembly in macrophages led to a decrease in the amounts of Lrp1 protein in macrophages, due to repressed translation. Inhibiting SG assembly by genetic knockout or a G3BP1-specific inhibitor alleviated AR symptoms with increased LPR1 protein levels and rescued efferocytosis in macrophages. This work provides evidence for a possible mechanism of the role for macrophages in AR and a potential treatment target.

Response: We appreciate your valuable feedback and evaluation of our work. In

response to the concerns you raised, we have now provided detailed point-by-point responses supported by new data. We believe that these revisions adequately address your concerns in the updated manuscript. Thank you very much for your consideration.

Major points

Question 1: From 2h, it seems the SG formation is indicated in Th2 priming as well as AR responses at the local tissue. Thus, whether the former is dampened, precluding subsequent changes in the local tissue becomes a key question. The authors should show that 1) after i.p., but prior to i.n. OVA-IgE titers are the same in ctrl and OVA-treated animals (same for S6G), and that 2) Th2 biasing with IL-4 + anti-CD3 or similar (in vitro) is unperturbed in the *G3BP1*^{mac-/-} line and *LRP1*^{mac-/-} line.

Response: Following your suggestions, we examined serum OVA-specific IgE levels in *G3bp1*^{mac-/-} and *Lrp1*^{mac-/-} mice after i.p. sensitization but prior to i.n. stimulation, and found that serum OVA-specific IgE levels were comparable between the floxed and knockout mice (**Figure 5a for Reviewer**). We also isolated naive CD4⁺ T cells from the spleens of *G3bp1*^{fl/fl} and *G3bp1*^{mac-/-}, as well as *Lrp1*^{fl/fl} and *Lrp1*^{mac-/-} mice using a magnetic-activated cell sorting kit (CD4⁺ T Cell Isolation Kit, Miltenyi Biotec), and these T cells were activated with plate-coated anti-CD3 Ab (2 µg/mL) and anti-CD28 Ab (2 µg/mL). After two days, the cells were supplemented with mouse IL-4 (20 ng/mL) and anti-IFN-γ Ab (10 µg/mL) to induce Th2 activation following previously established protocols (J. Clin. Invest. 2024, 134, e165689). We then measured *Il-4* mRNA expression using qRT-PCR and found no significant difference between the floxed and knockout mice (**Figure 5b for Reviewer**). Therefore, Th2 priming is not significantly influenced by the absence of *G3bp1* or *Lrp1* in macrophages.

Figure 5 for Reviewer. a. Serum OVA-specific IgE concentrations in *G3bp1*^{fl/fl} and

G3bp1^{mac-/-}, as well as *Lrp1^{ff}* and *Lrp1^{mac-/-}* mice after three times i.p. sensitization **but prior to** i.n. stimulation (n=3). **b.** *Il-4* mRNA expression levels in Th2 cells from *G3bp1^{ff}*, *G3bp1^{mac-/-}*, *Lrp1^{ff}* and *Lrp1^{mac-/-}* mice (n=3). Δ , not significant.

Question 2: Given that two MO targeting cre lines have been used to show a positive and negative impact on AR, the impact of the MO cre's on disease needs to be shown as null, or if the cre targeting alone modifies disease outcomes. That is, c.f. *Lyz2-cre*, *CSFR1cre* to wild type in at least one model.

Response: Following your insightful comments, we now show the AR symptoms data of OVA-induced AR in wildtype, *Lyz2-cre*, and *Csf1r-cre* mice, and there is no significant difference (**Figure 6 for Reviewer**).

Figure 6 for Reviewer. Times of sneezes and scratches in each mouse were counted for 15 min following the final i.n. challenge in OVA-induced AR model (n=5). Δ , not significant.

Question 3: 3f. Assay is crude and susceptible to the spurious population coming through at the lower end of the F4/80 gate. Please also report PEhi frequency by adjusting the PE+ gate to 5k on the PE-Jurkat axis. Similar for J, where part of the negative population is included in the gate. A no-jurkat control should be included in the assay, and ideally, unlabeled Jurkat cells or BMDE used. MFI of +ve fraction may further be a helpful metric in establishing veracity of outcome and sensitivity of assay to reveal differences.

Response: In response to your comments, we have now included the control groups in **new Supplementary Figure 3f**. PE high frequency of Figure 3f and j are now presented in **new Supplementary Figure 3g, m and o**. The MFI values of the PE⁺ fraction in Supplementary Figure 3g, m and o are also calculated (**new Supplementary Figure 3h, n and p**). Additionally, the flow cytometry analysis of no-Jurkat control and unlabeled Jurkat control are now included in the efferocytosis assay (**new Supplementary Fig.**

3f).

Question 4: Legends throughout should state statistical tests used in panels. E.g. S3F the test type is *impt* for small differences and needs to be stated. It should not be a *t*-test (as used throughout), approach should be revised after consulting a statistician. This may mean the data need normalization and then combining across the multiple experiments for robust interpretation, or the conclusions reframed. Also need to state biological replicate # vs technical replicates as not clear currently throughout.

Response: Following your comments, we have consulted a statistical expert and reanalyzed all the experiment data. Consequently, the p-values have been revised accordingly. Additionally, the updated figure legends now include details on replicates and n values, specifying whether they are biological or technical replicates.

Question 5: Fig S2b,c. The specificity to myeloid lineage should be demonstrated. Show a non-macrophage cell type (e.g. T cells) for *G3bp1* expression.

Response: We isolated CD3⁺ T cells from the spleens of *G3bp1^{fl/fl}* and *G3bp1^{mac-/-}* mice using a magnetic-activated cell sorting kit (EasySep™ Mouse T Cell Isolation Kit, Stem Cell Technologies). *G3bp1* expression levels at both mRNA and protein were examined, and they were not significantly influenced by *G3bp1^{mac-/-}* (new Supplementary Fig. 2b and c), presenting the specificity of macrophage-specific knockout.

Question 6: Fig S2F. The vehicle control is missing. Should have the vehicle treated control to show that there is increase AR symptoms.

Response: The vehicle controls are now presented (new Supplementary Figure 2f).

Question 7: Figure 4 title states that SG assembly internalizes *Lrp1* mRNA to suppress its translation. Stronger evidence for this is needed to make this claim as it could equally be enhanced protein degradation. While Fig S5c/d speak to no change in degradation, it should be verified by FACS as in 4g,h and presented in the supplement alongside the Western blots. A confirmation that translation was inhibited with MG132 and CQ would also strengthen the claim.

Response: Following your insightful comments, the expression of LRP1 in macrophages, pretreated with MG132 or CQ, are now measured using flow cytometry

in HDM-stimulated PMs and G3BP1-deficient PMs (**new Supplementary Figure 5d and f**), confirming that the inhibition of LRP1 protein levels by G3BP1 and stress granule assembly is not due to modulation of LRP1 degradation.

Question 8: *Fig 6h, the baseline sneeze and scratch responses in vehicle treated mice across the strains should be presented. If zero, this should be stated.*

Response: The baseline sneeze and scratch in vehicle controls across the strains of Figure 6h are now presented in **new Supplementary Figure 6i**.

Question 9: *Fig 7G. The control should include CD68+ cells; at the moment it is simply showing a lack of MO in the control, which is different from SG formation in the disease state specifically. Further, given this is the human link, 7G should be quantified i.e. SG/mm2 or similar in the two contexts.*

Response: Following your insightful comments, the areas containing CD68⁺ cells from the healthy controls are now presented in **new Figure 7g**. The numbers of CD68⁺ cells and CD68⁺ cells with SG assembly per 10 random fields are now quantified (**new Supplementary Figure 7j and k**).

Minor points

Question 1: *The introduction should be simplified. It should state the rationale for the study, but at the moment is quite leading towards the outcome. Paragraph 1 and 2 could be integrated, and the leading sentence L105-106 should remove 'we presume' and be rewritten more agnostically. The manuscript would benefit from parsing by an English editing service.*

Response: In accordance with your instructions, we have restructured the Introduction section. Additionally, we have utilized a professional English editing service to refine our manuscript, in order to meet the required standards.

Question 2: *Fig 1a The G3BP1 imaging was with poor quality for the identification of SG assembly. Some means of quantifying granules in a field may make this more robust (SG/mm2 or similar).*

Response: Following your comments, we have now quantified the number of SG aggregates per 10 random fields (**new Supplementary Figure 1a**).

Question 3: *For the mouse models and measurements of sneezing/scratching, more detail is required. That is, clarify whether or not the mice were anaesthetized for i.n.s.*

Response: The mice were not subjected to anesthesia during i.n.s. and the measurement of their AR symptoms. Further details are now incorporated in the AR mouse models of Methods section.

Question 4: *I couldn't find "extended Fig 4A-C". Should this be "supplementary Fig 4A-C"?*

Response: This error is now corrected in the revised manuscript.

Question 5: *L305: For the DKO, what is the cre driver? Both or just one. Not clear from text or methods.*

Response: DKO mice were generated by crossing *G3bp1^{ff}Lrp1^{ff}* with *Csf1r-Cre* mice, as *Lrp1* and *Lyz2-Cre* localize on the same chromosome, this information is now included in the revised maintext (Page 14).

Question 6: *What are the invading cells evidenced in 2B in the G3PB1Mac^{-/-} line? Everything Th2 associated in the mice is down in the rest of the figure, so what goes up to explain the increase in nuclei needs to be demonstrated. For e.g. are they CD45+ cells?*

Response: For each mouse in the histochemical analysis, individual segments are examined, and cell densities may be different because of the possible distinct anatomical positions. As H&E staining primarily reveals the mucosal damage of mice, we present the images to show the integrity of nasal mucosa and then quantify it using a scoring system (**new Supplementary Figure 2i**).

Importantly, equal-weight nasal mucosa tissues were collected from both *G3bp1^{ff}* and *G3bp1^{mac^{-/-}}* mice and then analyzed using flow cytometry, which provides a more precise quantification of immune cell infiltration. No significant difference was observed in the total cell counts between the two strains. However, the proportion of CD45⁺ cells in the nasal mucosa of *G3bp1^{mac^{-/-}}* mice was lower compared to that in *G3bp1^{ff}* mice (**Figure 7 for Reviewer**).

Figure 7 for Reviewer. Flow cytometry of CD45⁺ cells in the nasal mucosa of control and AR mice (n=3). *, p<0.05; **, p<0.01, Δ, not significant.

To Reviewer #1:

Remarks to the Author:

All my issues were addressed adequately. I recommend this work for publication.

Response: Thank you very much for your approval, and we deeply appreciate your recognition.

To Reviewer #2:

Most of my concerns were addressed.

Response: Thank you very much for your approval, and we deeply appreciate your recognition.

Remarks to the Author:

Question 1: *However, the results in supplemental figure 2l are very important and should be moved to figure 2.*

Response: Following your comments, we have now moved the results of previous **Supplementary Figure 2l to Figure 2f.**

Question 2: *Also, the increased expression of IL-10 in *g3bpMac*^{-/-} macrophages is important and should be examined in isolated nasal/peritoneal macrophages and tissue samples from nasal mucosa from *g3bpMac*^{-/-} mice during disease. The text should also indicate these findings.*

Response: In response to your comments, we have now isolated nasal mucosa tissues and macrophages (NMMs) from OVA and HDM-induced AR mice, and examined the expression level of *Il-10* mRNA. Our results demonstrate that G3BP1 deficiency in macrophages leads to the increased IL-10 expression (**new Figure 2g and 2i**). Additionally, the increased IL-10 expression was also determined in G3BP1 deficient peritoneal macrophages, especially after efferocytosis (**new Supplementary Figure 3q**). These contents are now incorporated into the revised manuscript accordingly.

Question 3: *The abstract should be amended to “suppressed the clearance of apoptotic immune cells resulting in reduced Mres generation and IL-10 production...”, Moreover, the sentence “Next, less cytokines expression in the nasal mucosa (Fig. 2f and Supplementary Fig. 2m), and lower cytokines production in NLF of G3bp1mac^{-/-} mice than those of controls were determined (Fig. 2g)” should be clarified and clearly indicate that the differences in IL-4, IL-5, IL-13, IL-6, Prg2, Epx, and Mpo (mRNA or protein) were found only in challenged mice and not their unchallenged controls.*

Response: Following your insightful comments, these contents have now been revised accordingly in the updated version of our manuscript.

To Reviewer #3:

Remarks to the Author:

The response letter addresses concerns raised. To improve the manuscript quality in a further revision round, i) all data supplements should be referenced in the body text, and ii) incorporating the Th2 differentiation data presented in the response letter but not currently in the main document into a supplementary would improve veracity.

Response: Thank you very much for your approval, and we deeply appreciate your recognition. Following your comments, we have now checked all data supplements referenced in the body text, and have incorporated the Th2 differentiation data in **new Supplementary Figure 2o** and **Supplementary Figure 6g**.

To Reviewer #4:

Remarks to the Author:

Response: Thank you very much for your approval, and we deeply appreciate your recognition.